# Stability and Performance Enhancement of Perovskite Solar Cells: A Review

**Maria Khalid and Tapas Kumar Mallick \***

Environment and Sustainability Institute, University of Exeter Penryn Campus, Penryn TR10 9FE, Cornwall, UK; bintekhalid29@gmail.com
* Correspondence: t.k.mallick@exeter.ac.uk

**Abstract:** Perovskite solar cells (PSCs) have seen a rapid increase in power conversion efficiencies (PCEs) over just a few years and are already competing against other photovoltaic (PV) technologies. The PCE of hybrid PSCs exhibiting distinct properties has increased from 3.8% in 2009 to ≈30% in 2023, making it a strong contender for the next generation of PV devices. However, their long-term stability is a critical issue that must be addressed before these devices can be commercialised. This review begins with a discussion of the evolution of different generations of solar cells, and the following part presents details of perovskite characteristics and prospective strategies to improve their performance. Next, the relationship of stability of PSCs with different environmental conditions, including moisture, UV light, and temperature, is discussed. Besides the development of PSC–silicon tandem solar cells, an efficient way to improve PCE is also discussed. Towards the end, we discuss a novel idea of implementing PSCs with a concentrated PV application in order to achieve higher efficiency and compete with other PV technologies by catching incident high-proton density. This review offers perspectives on the future development of emerging PSC technologies in terms of device performance enhancement and improved stability, which are central to tandem and concentrated PSC technology.

**Keywords:** photovoltaic; perovskite solar cell; efficiency; stability; concentrated optics

## 1. Introduction

Perovskite solar cells (PSCs) have the most significant improvement in terms of efficiency in recent years. Perovskite is an organic–inorganic hybrid compound with an $ABX_3$ crystal structure. The fundamental crystal structure of perovskite follows a $BaTiO_3$-type. In the structure, X represents halogens such as $I^-$, $Br^-$, $Cl^-$, or oxygen, whereas A and B are cations of different sizes, having 12 and 6 coordinates with X anions, respectively. For cation A, i.e., methylammonium, formamidinium does not directly contribute to valence band maxima and conduction band minima. However, it does affect the lattice constants, and it was found that bandgap increases with increasing lattice parameters. The ideal structure of the perovskite is similar to a body-centred cubic structure with additional anions on the faces of the unit cell. The perovskite absorber and kind of PSC device architecture benefits from a high charge collection efficiency and low recombination of carriers, which are indispensable to realising high-power conversion efficiency (PCE). Thus, enhancing the material quality of the perovskite absorber is necessary to increase the electron mobility and lifetime of carriers and decrease the defect density.

In 2009, Japanese scientists observed that organic metal halide perovskites ($CH_3NH_3PbI_3$) are quite like dyes that are capable of absorbing sunlight. The perovskite material was employed as a sensitiser in a dye-sensitised solar cell (DSSC) and obtained a power conversion efficiency (PCE) of 3.8% in the presence of a liquid electrolyte [1]. Later on, in 2012, for the first time, 9.7% PCE was reported [2]. This disruptive finding quickly led to perovskite's emergence in the solar cell industry. Perovskite has drawn considerable attention and



interest of many researchers due to its ease of fabrication, low non-radiative carrier recombination, and high absorption rate. Moreover, there are some other appreciable features such as long carrier diffusion length; charge transport; reduced recombination; high carrier mobility; and the formation of free charge carriers at the surface of the perovskite absorber material, which results in a useful collection of fill factor (FF) and open-circuit voltage (Voc), as well as a high internal quantum efficiency close to 100% [3]. All these factors together allow the perovskite material to provide a higher PCE record beyond 20%.

In February 2022, another spin-coated cell with improved efficiency of about 20.1% and 84.3% of FF was reported. The improvement was driven mainly due to higher film quality and perovskite absorber film. The highly oriented perovskite film with low defects intensifies the absorption rate, photocarrier transportation, and extraction, leading to promising FF output. This strategy has been suggested to be used to further develop PSC to the industrial level with different compositions [4].

Later, the highest certified PCE reached 25.2%, confirmed by the international authority and authenticating National Renewable Energy Laboratory NREL in 2021 [5]. To date, scientists have broken the efficiency record for multi-junction perovskite cells to about 30%, leaving little room for more growth. However, solving the problem of transferring high efficiency from laboratory small-area devices to large-area perovskite modules is a vital challenge. Since the maximum theoretical PCE (Shockley–Queisser limit) of the PSCs employing $CH_3NH_3PbI_{3-x}Cl_x$ is 31.4%, there is still enough space for development. Moreover, the Shockley–Queisser limit could be attained with a 200 nm thick perovskite solar cell by integrating a wavelength-dependent angular restriction design with a textured light-trapping structure [6]. The possible reason for this small fraction is the instability of perovskite materials and other associated problems, which will be discussed in the following sections. In 2018, certified PCE for 1 cm$^2$ size Si/perovskite tandem solar cells was 27.3%, as reported by Oxford photovoltaic. However, Helmholtz–Zenturm Berlin recently surpassed 29.15% of PCE a year later. It is noteworthy that hybrid PSCs have developed to higher efficiencies in a few years as opposed to conventional Si solar cells. In September 2022, another result of the perovskite solar cell reported by NREL with more excellent stability and unique structure was presented. The certified PCE of 24% with stabilisation of 87% after exposure of 2400 h under 1 sun illumination was achieved [7]. The highest ever certified efficiency of a perovskite solar cell by an accredited laboratory (Newport, USA) was 27.53%. The measured values of Isc, Voc, and FF were 25.80 mAm$^{-2}$, 1.179 V, and 84%, respectively [8]. The encapsulated device without a UV filter cut-off retained approximately 88% of its initial efficiency after an exposure of 600 h to full sunlight illumination.

The emergence of perovskite in the last ten years with some milestones is described in Figure 1. PSC appeared as a top breakthrough in the solar technology era, showing an incredible increment in PCE from just 3.8% in 2009 to almost 30% in 2022.

The potential of PSC during the decade of technological advancements has reported several milestones. In 2021, Korean researchers obtained a PCE of 25.8% for a single-junction cell with an interlayer between ETL and perovskite, eliminating the need to produce minimised interface defects, ultimately producing minimised defects interfaces. These cells retained 90% initial efficiency after 500 h of illumination exposure. This architecture was certified by the U.S. Department of Energy NREL. An certified efficiency of 25.6% for single-junction PSC was achieved by EPFL, the energy institute in Switzerland [9]. These results disabled the previous records of 25.2% PCE obtained by scientists at the Massachusetts Institute of Technology and 25.17% achieved by UNIST researchers [10].

An innovative idea to achieve high efficiency with low cost using concentrated light has appeared as an effective route in the PV era. Several technologies have been put forward to enhance the efficiency of PSC.

Wang et al. experimented with assembling concentrated optics with PSC devices. The experiment was performed under 14 suns and reported the achievement of an efficiency of 23.6% [11].

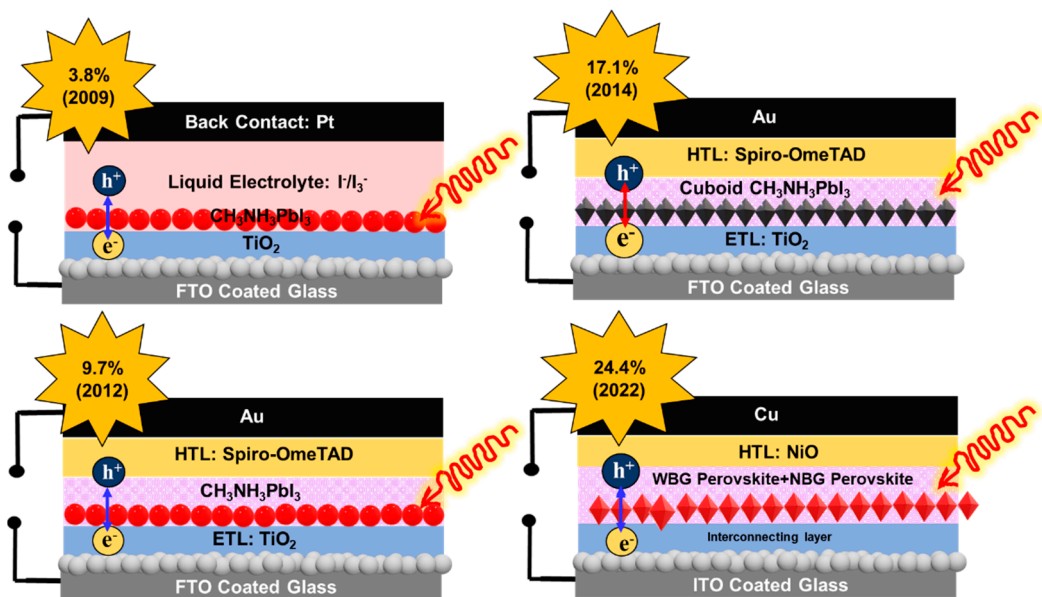

**Figure 1.** Emergence of perovskite solar cells over 13 years.

Timeline of the improvement of PSC and the foreseeing of further development is shown in Figure 2.

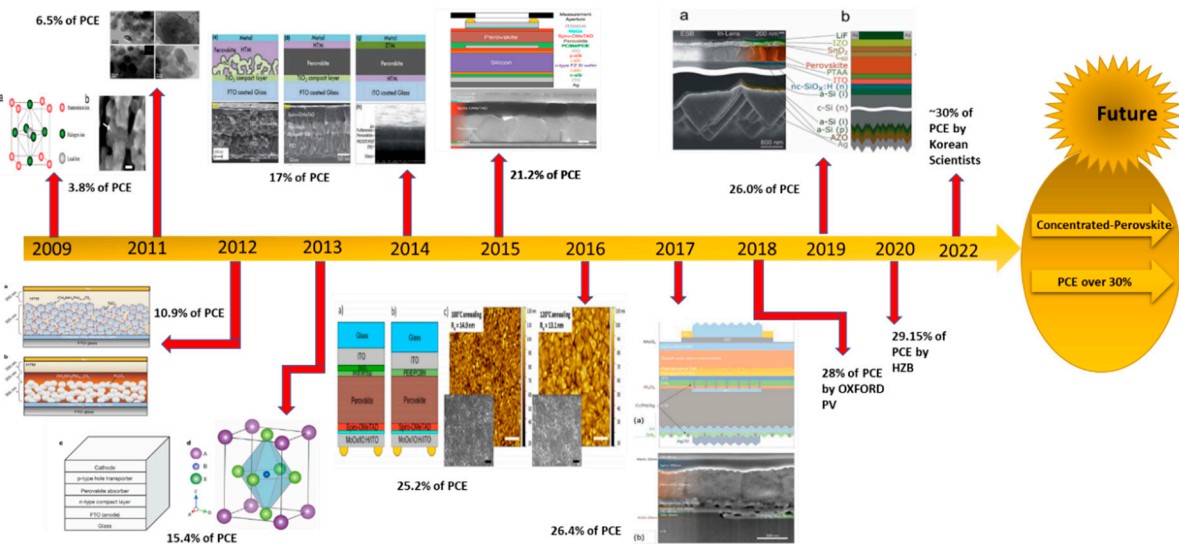

**Figure 2.** Timeline of important breakthroughs in perovskite solar cells.

Some associated problems that can cause low performance of PSC systems can be the material's toxicity [12]. Lead has been used as a significant constituent of almost all competently performing PSCs to date, potentially raising toxicity issues during the preparation process of the device, as well as during disposal. Due to the use of lead, they also gradually degrade when exposed to air, moisture, or ultraviolet radiation. To circumvent the toxicity issue, several researchers use Sn rather than Pb to cope with this problem [13,14]. Many research efforts are still in progress to address the significant challenge of the stability of perovskite materials in the ambient environment by the interface of new device architecture and engineering.

In the present paper, a topical review of advances in the PSC industry is discussed. It is, however, by no means an exhaustive claim, but it is different from other reviews as it covers a variety of topics, namely, evolution; architecture development; the basics of working a PSC device; stability; the hysteresis curve; and degradation mechanisms due to

different factors, e.g., the environment, moisture, and humidity, and it offers the new idea of using perovskite solar cells with the implementation of the concentrated optic.

## 2. Evolution of the Perovskite Solar Cell

The perovskite solar cell consists of perovskite, classed as an inorganic–organic hybrid compound that harnesses solar energy and works as a charge carrier conductor. A large revolution in perovskite materials in solar cell technology in terms of providing extraordinarily effective and phenomenally improved power conversion efficiency has been witnessed since 2009. The reports of authors dating back to the early 1990s show the inception of studying perovskite as a solar absorber.

In 2006, Japanese researchers achieved an exceptional PCE of 2.2%, which is regarded as a landmark. Dye-sensitised solar cells with $CH_3NH_3PbBr_3$ as a sanitiser was used in the experiment [15]. The first organic halide compound reported in 2009 was fabricated using $CH_3NH_3PbBr_3$ as a sanitiser. They recorded a PCE of 3.81% for the $CH_3NH_3PbBr_3$ device and 3.13% for $CH_3NH_3PbI_3$. Later, the same group again measured a PCE of 6.5% using the $CH_3NH_3PbI_3$ compound with iodine liquid electrolyte contact and a better fabrication condition. Figure 3 shows the PSC's evolution timeline and future development aspects.

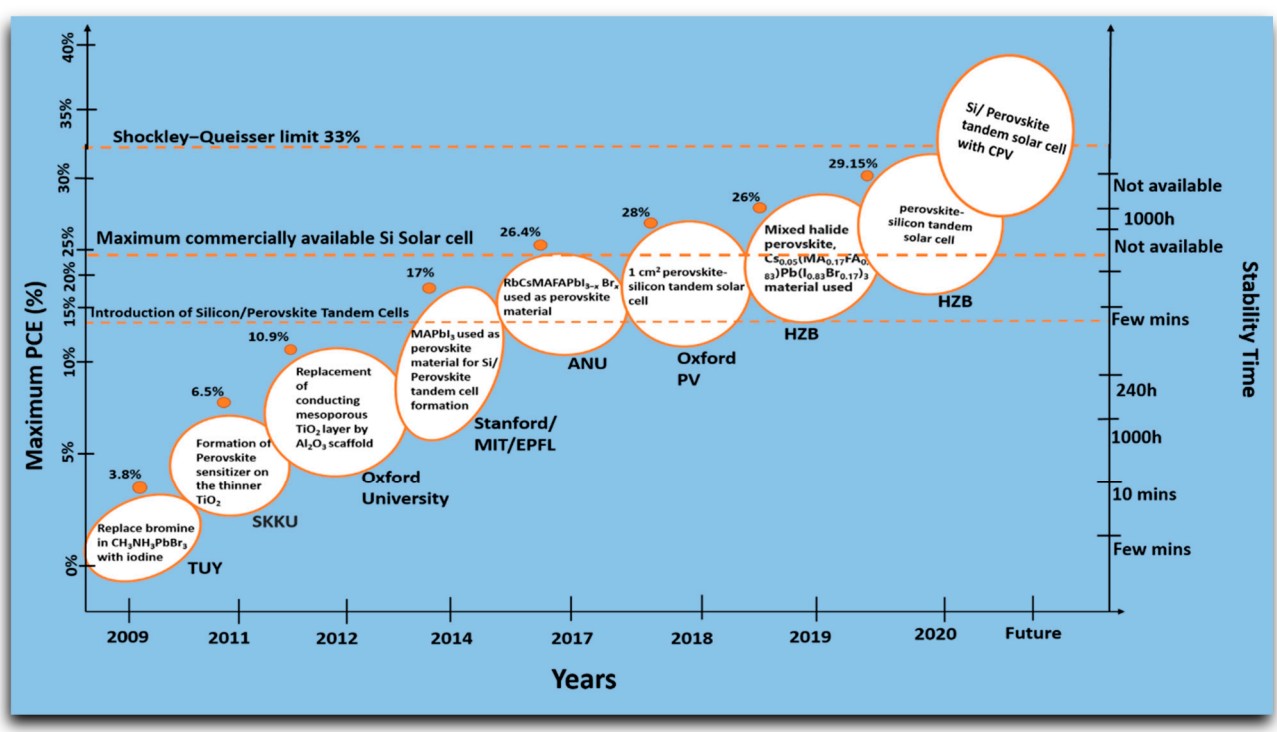

**Figure 3.** Timeline of the development of PSCs with stability time and the reported efficiency and future foresight.

However, this type of PSC was unstable due to the decomposition of perovskite nanocrystals in the iodine liquid electrolyte. Afterwards, numerous researchers entered this field in the following years to improve the PCE of perovskite solar cells. The fundamental photophysical properties, functioning, and unique nature offer many advances in optoelectronic applications that triggered this technology [16]. Perovskite materials also have distinct electronic properties such as piezoelectric properties, thermoelectric properties, superconductivity, and semi-conductivity, which are dependent on the selected materials. Due to these favourable properties, exciting experimental results, the mechanism to explain the higher PCE of these materials, various fabrication methods, and a considerable increase in efficiency beyond 25% were reported only in a few years. The PCE of 9.7% was achieved using Spiro-OMeTAD as the hole transport material (HTM) and

$CH_3NH_3PbI_3$ as the perovskite solar absorber to fabricate solar state DSSCs [17]. This DSSC has shown dramatically improved device stability compared to liquid electrolytes, yet the device stability is a significant challenge for researchers in terms of commercialisation and mass production.

Figure 4 shows different problems associated with PSC to affect its output, including the toxicity of the material, instability of different parameters, light entrapment, and degradation mechanism. The toxicity of Pb can be resolved by substituting appropriate materials such as bismuth (Bi) and tin (Sn), or by developing a green synthesis of the toxicity-free material. A light trapping mechanism in PSC can also cause perovskite crystal structure, architecture, and band gap degradation. In contrast, the formation of metal halide and layer degradation can occur within the fabrication process. Moreover, phase transition can also occur due to the isotropic nature of the perovskite material, leading to degradation. The stability factors are discussed in detail in the following sections.

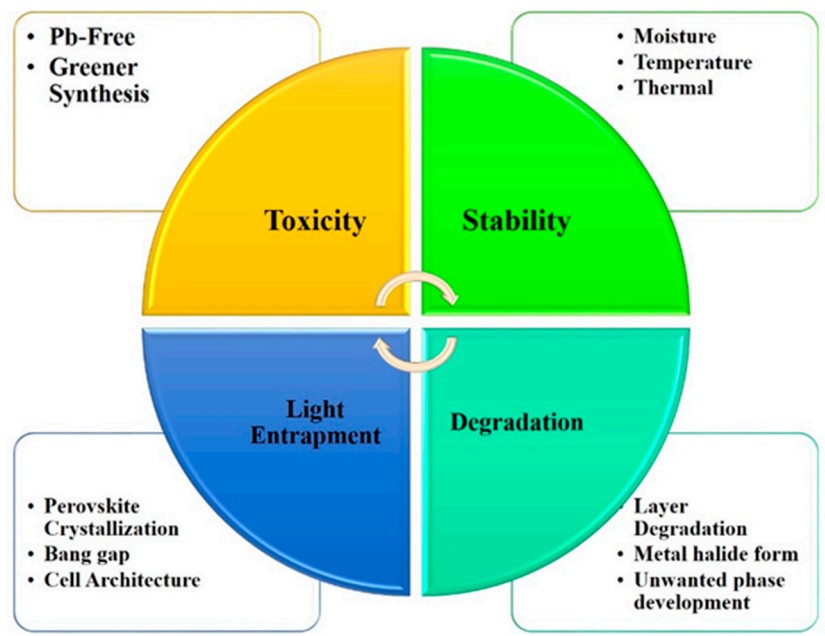

**Figure 4.** Various associated problems with perovskite solar cells.

### 2.1. Characteristics of Perovskite Solar Cells

The optical properties of PSC, including bandgap, polarisation, absorption coefficient, and photoluminescence, have been discussed widely in the literature. Regarding the optical properties of perovskite materials, e.g., $CH_3NH_3PbI_3$ and $CHNH_2PbI_3$, they manifest relatively higher absorption coefficients than Sn-based perovskite materials. Replacement of methylammonium (MA) with other organic cations also allows for the tuning of the bandgap materials without affecting valance band maximum (VBM) by modifying the Pb-I-Pb bond length and angle [18].

The optical behaviour was observed to investigate the variation in energy levels and bandgap of the perovskite materials. The reduction was detected in the valence band maximum (VBM) and conduction band minimum (CBM) levels to 110 meV and 77 meV, respectively, by the increment of temperature to 25–85 °C by employing absorbance and PL spectroscopy and the further increase of 33 meV in the bandgap level of $MAPbI_3$ with the rise of temperature [19]. The researchers used the density functional theory (DFT) to present a more in-depth perception of thermal expansion that was dependent on a shift in VBM level, providing significant results for further designing $MAPbI_3$ at different applied temperatures. Another group observed an 100 °C temperature causing a shift in the band edge of $MAPbI_3$. The decomposition of $MAPbI_3$ at a lower temperature cannot be used commercially [20].

Single-crystal $BaZrO_3$ showed an indirect and high optical band gap energy of $\approx$4.8 eV. The Raman study exploited the second-order spectrum and opened up the possibility for fundamental studies due to the availability of single-crystal $BaZrO_3$ [21]. The plot of spectra at ambient temperature and low temperature did not indicate any phase transition. Slight changes could have been due to thermal effects.

Another valuable optical property of PV material is photoluminance (PL), which helps to provide a wide range of information about bandgap, charge separation, and chemical purity. The phenomena behind photoluminance are that when light is directed onto semiconductor material, protons absorption occurs on the material surface, and photo excitement occurs, followed by relaxing excitation to the ground state. This phenomenon was investigated by using organometal halide. An integrating sphere is suggested to reduce angular distribution for a better PL effect on PV cells. Some substituting methods are also recommended, e.g., substitution of methylammonium ($\lambda$ = 776 nm) and formamidinium ($\lambda$ = 776 nm) in lead-iodide-based perovskite [22]. The formation of broader PL peaks was reported by an increased formamidinium ratio, indicating the evolution of the reliable solution of MA and FA in the perovskite lattice. The investigation of the simultaneous intercalation of MA and FA cations with different compositions by XRD has been reported.

The same phenomena were observed for substituting iodide with bromide in lead-halide-based perovskite [23]. The property of PL in perovskite material is highly stable and showed promising results when exposed to various environments [24]. The charge separation mechanism can be figured out via PL quenching measurements. The PL behaviour of $CH_3NH_3PbI_3$/SpiroMeTAD and $CH_3NH_3PbI_{3-x}$ $Cl_x$ /PCBM was studied [25,26]. A significant decrease was noticed in PL intensity when both materials were in contact with the electron transport layer (ETL) and the hole transport layer (HTL) caused by the injection of electrons and holes. Due to the high conduction band, electrons are injected into the mesoporous layer but not the $Al_2O_3$ layer. Some studies have reported that inorganic HTL, e.g., CuSCN and NiO, act as charge separators. However, NiO showed less efficiency in terms of charge separation than CuSCN [27].

The impedance measurement is a crucial parameter in perovskite solar cells in terms of providing information about carrier transport, diffusion length, and recombination phenomena. According to impedance studies, the carrier conductivity for two different mesoporous and planar devices was similar, but the planar-based cell had higher recombination resistance than mesoporous-based cells [27]. Impedance studies also provide information about the occurrence of polarisation in PSC devices. However, perovskite material polarisation phenomena are likely to cause I-V hysteresis [28].

Electronegativity as a fundamental chemical property of PSC may explain the competence of an ion to attract electron density in the chemical bond. In halide ions, an electronegativity rise from $I^-$ to $F^-$ originates from the cutback of ionic size. However, $F^-$ exhibits excellent robust bonding with the H atom of A site and B site compared to other halogens. The formation of better charge transport leads to an enhanced electron mobility rate and also limits lead leaching. The B-X bond can lead the way to a lower bandgap, which is requestioned for Vis and NIR range of optoelectronics [29].

PSC performance and different associated factors are shown in Figure 5.

Depending on the unique characteristics of PSC, different strategies have been offered to improve the PCE of the PV systems.

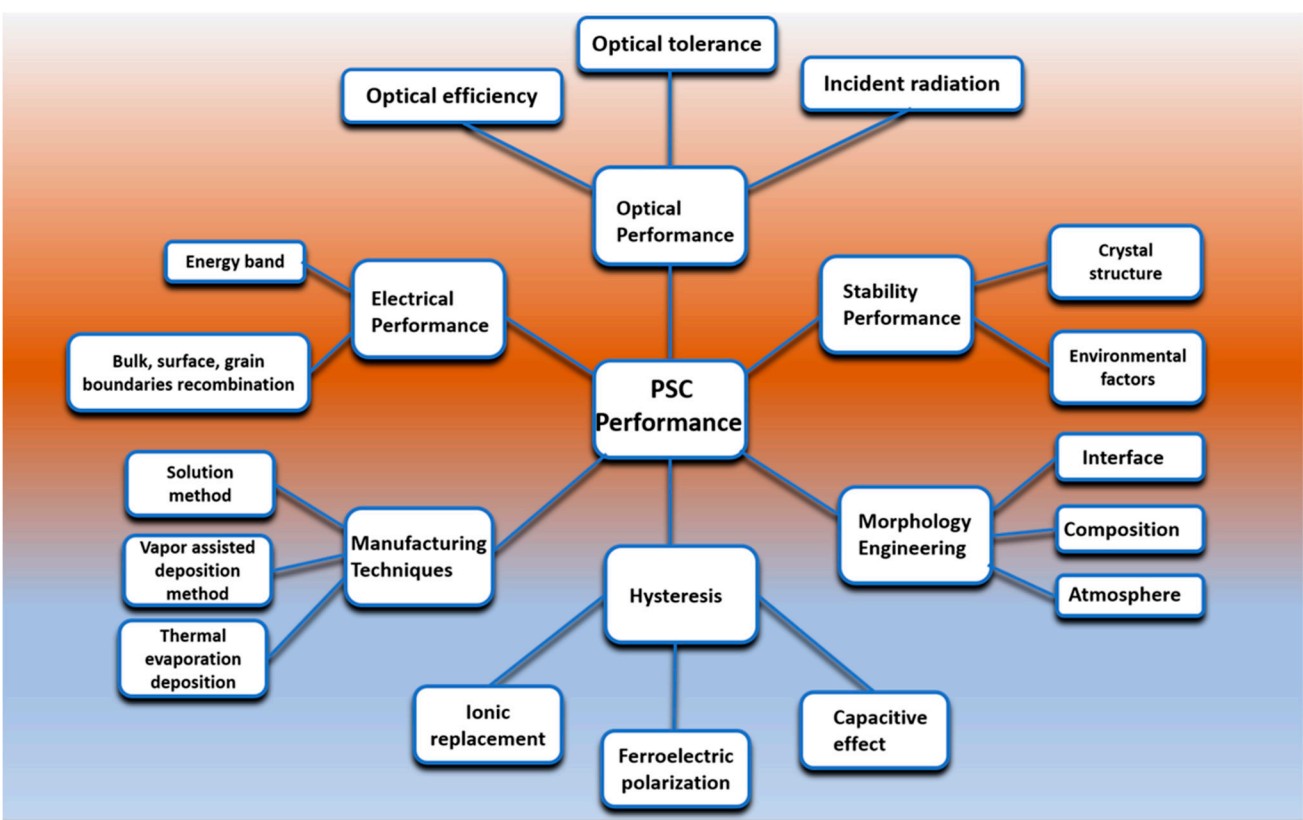

**Figure 5.** Dissemination chart of PSC performance and dependence on associated factors.

### 2.2. Improving Efficiency through Structure Modulation

Perovskite materials play an important role among other components of perovskite solar cells for a better absorption rate and photoelectric conversion. Optimised hybrid PSC systems show better crystallinity and quality with 17% of PCE [30]. The most studied perovskites for PV devices are $CH_3NH_3PbI_{3-x}$, $Cl_X$, $CH_3NH_3PbBr_3$, $CH_3NH_3PbI_3$, $HC(NH_2)_2$, $Pb(I_{1-x}Br_x)_3$, $CH_3NH_3Pb(I_{1-x}Br_x)_3$, and $CH_3NH_3SnI_3$ [31]. Perovskite compositions with simple cations and anions by employing at each site of A-, B-, and X-sites as $ABX_3$ are signified as "simple perovskite" and have been under consideration for the investigation of various characteristic features and for the improvement of the PCE of PSC [32]. Optimising material and structure is the keystone to enhancing the photocurrent conversion efficiency of perovskite devices.

Due to the molecular matrix of methylammonium concerning the crystal axis, an exciting phenomenon occurs; they undergo structure phase transition with temperature variation. The crystal structure of $MAPbX_3$ was examined with the produced sample. As a result, cubic, tetragonal orthorhombic phase transitions were formed at low temperatures due to the high order of organic cations with an C-N axis through the quenching of the molecular matrix [33]. The distribution of MA ions has influenced Pb-I bond length, Pb-I-Pb bond angle, and bandgap energy. Still, no direct effect on valance band maximum (VBM) and conduction band maximum (CBM) indicates the critical role of MA ions in both the electron structure and geometry stability.

In the ideal perovskite structure composite of $ABX_3$, A and B are cations with different oxidation numbers, whereas X is an anion. The large A cations usually occupy the body centre position of the cubic structure, and the B cation is arranged at the corner. In contrast, the X anion is arranged on the edges of the structure. The perovskite material has an isotropic nature and can occupy equivalent directions as 6 for trigonal, 12 for orthorhombic, and 8 for rhombohedral.

### 2.2.1. A-Site Cation

In the crystal structure of perovskite, it is generally understood that the A-site cation has no direct impact on electronic properties, but the bandgap structure can be tuned to adjust, which can affect the electronic characteristics of the perovskite material. A cation can expand or contract the lattice if it is of a smaller or larger size that can affect the bandgap by changing the B-X bond length [34]. The first mixed A-cation $MA_xFA_{1-x}PbI_3$ was reported to tune the bandgap by varying ratios of $MA^+$ to $FA^+$ and attained an efficiency of 14.9% [22]. The enhanced PCE of this perovskite material compared to the $MA^+$ engaged to the higher absorption rate due to better film quality, fewer pin-holes, higher crystal quality, larger grain sizes, and smaller roughness within the grains.

Another stacked structure of $FAPbI_3/MAPbI_3$ films by ion exchanging was prepared, which increased the absorption rate and higher efficiency of 16.01% with a current density of 20.22 mA cm$^{-2}$ [35]. The three-dimensional symmetrical structure was also reported by employing the vacancy at the A site by a tiny monovalent cation, e.g., rubidium (Rb), caesium (Cs), farmamidium (FA), and methylammonium (MA). The most used cation in mixed A-site perovskite is methylammonium (MA). For $CH_3NH_3PbI_3$ devices, power conversion efficiency was reported beyond 15%. The structure of $CH_3NH_3PbI_3$ is tetragonal symmetry rather than cubic due to the tiny dimensions of the $CH_3NH_3^+$ cation. As a result, it exhibits an absorption rate of 1.51–1.55 ev, which is significantly higher than the optimised limit of Shockley–Queisser for mono junction devices [36,37]. It is well known that inorganic-material-based perovskite devices show better efficiency, performance, and stability as compared to organic material. The idea of replacing organic cations by inorganic cations to design PSC devices was initiated by Choi et al. $Cs_x(MA)_{1-x}PbI_3$ PSC was devised, and it obtained 7.68% efficiency [38].

Formamidinium (FA) has been extensively probed due to its favourable characteristics of better symmetry than $CH_3NH_3^+$. A bandgap of $\approx$1.43–1.48 eV with an absorption rate of 840 nm, close to the optimum value of 1.4 ev that was reported by the $FAPbI_3$ crystal process [36]. A non-perovskite polymorph of $FAPbI_3$ was devised, and it was anticipated to limit PV efficiency due to unpropitious bond collaboration with the $TiO_2$ layer [39]. In future, the complete elimination of non-perovskite is expected to lead to the efficiency of FA-based devices to exceed that of MA.

Recently, the $Rb^+$ cation with a 0.152 nm ionic radius received extensive attention due to the probability of the Rb-mixed PSC to enhance both the efficiency and stability of the devices. Park et al. exploited an $(FA/Rb)PbI_3$ device using a 5% Rb quantity and reported 16.15% and 16.2% of PCE [40]. $CsPbI_3$ quantum dots were used, and the film was exposed for 60 days in an ambient environment and achieved efficiency up to 10.77% with a Voc of 1.23 V [41]. This work showed high stability for PSC but still holds an organic HTM layer. Later, all inorganic devices without organic HTM and metal electrodes were presented to fabricate PSC with a coating of carbon on the $CsPbBr_3$ layer [42]. This kind of structure is promising in that it has the advantage of processing at an ambient environment, even without controlling for humidity.

### 2.2.2. B-Site Cation

The position of the B-site cation in the structure of hybrid perovskite material is usually taken by metals of the IVA group in a +2-oxidation state. The most widely used metals are $P^{2+}$, $Ge^{2+}$, and $Sn^{2+}$. Sn and Pb both belong to the same group in the periodic table, as mentioned above. Pb has a toxicity problem with it. Different alternatives of Pb have been reported, such as Sn in the fabrication of PSC devices [43]. Lead is the most studied material with higher performance, but lead has a toxic nature, which results in the instability of the devices [44]. Sn has less of an inert electron pair effect, and the toxic character of this metal also improves results in a reduction in the bandgap. Generally, $Sn^{2+}$-metal-based perovskite devices present a lower bandgap as compared to $Pb^{2+}$-based devices influencing the stability of the device and reduced PCE. Thus, the idea to use mixed Sn and Pb to prepare perovskite material in the B site was proposed to obtain an absorption near the

infrared region. It is believed hypothetically that MASnX$_3$ manifests extended and higher bandgap values than MAPbX$_3$ [45]. Zuo et al. reported an inverted planar structure device using a combination of double Pb-Sn perovskite and obtained 10.1% of PCE [46]. The same structure was used to fabricate PSC with modified C$_{60}$ Sn-Pb perovskite films and showed a PCE of 13.9%.

Moreover, when the compound of Sn-Pb perovskite with a C$_{60}$ additive was exposed to an ambient environment without any encapsulating, it showed excellent stability and superb efficiency [47]. Stoumpos et al. observed that both MASnI$_3$ and MAPbI$_3$ follow tetragonal arrangements in the ambient environment. Pb was replaced by Sn to produce a more uniform perovskite absorbing layer with enhanced co-ordination involving complexes [48,49].

Marshell et al. used inorganic CsSnI$_3$ perovskite material to elucidate the significant effect of adding SnCl$_2$ in the perovskite light-absorbing layer to stabilise the device without compromising on PCE [50]. Furthermore, a wide range of studies on replacing lead (Pb) by Ge or Bi has been reported to indicate the reason for PCE in electron–hole carrier recombination and solubility in perovskite material [51].

### 2.2.3. X-Site Anion

A handful of studies have been reported to explain mixed halide ions on the X site incorporated in perovskite material [52,53]. The electronic structure of perovskite ABX$_3$ is generally associated with the p orbit of X and B. The bandgap of perovskite material can be swayed by modifying the p orbit of the mixed X-site anion and for the absorption of visible light when exposed to sun radiation [54]. Noh et al. first demonstrated the PV properties of a mixed X halide anion, and this achieved 12.3% efficiency [55]. They examined both lower (<10%) and higher (>20%) Br content. They yielded high initial efficiency for lower content due to a narrow bandgap, but for higher content, they obtained excellent stability against humidity (RH 55% for 20 days). This work suggested that a close stack perovskite structure can offer prevention from the degradation of CH$_3$NH$_3$H$^+$$_3$.

Bromide is a favourable metal used to attain a high bandgap of perovskite, as has been reported earlier. Moscon et al. showed that assimilating Br metal in iodine-based perovskite can cause structural deformity, resulting in a higher bandgap [56]. MAPbI$_x$Br$_{3-x}$ was reported by chemical amalgamation to improve stability by tuning the bandgap, as well as improved PCE [55]. Chlorine has also been extensively studied in the literature to enhance the efficiency of perovskites. Cl presented excellent performance, mainly in planar heterojunction configuration, in order to improve carrier lifetime and diffusion length, and thus a perovskite device showed improved PCE. Cl subliming transformation into pure MAPbI$_3$ was suggested by Unger and co-workers. The electronegative of I$^-$ (2.66 EN) is close to Pb(2.33 En), so its bond character is neither covalent nor ionic but somewhat mixed. Thus, it showed the most stable behaviour when incorporated with perovskites. However, a downside of iodine-based hybrid halide perovskite is its instability towards humidity.

Consequently, further research needs to find the substitution of iodine or other mixed halide perovskites. The fluorine ion (F) as a worthy candidate for PV solar device application is emerging as a perovskite material due to its superior characteristics of electron withdrawing. Moreover, it has the potential of forming a hydrolytic bond (N-H-F) in comparison with other halides [57]. However, fluorine has the drawback of being tinier in size than iodine, which leads to an unfavourable tolerance factor and a considerable amount of strain lattice that can halt the capacity of light absorptivity and conductivity. Even though PSCs have improved their output using different techniques and materials, stability is still a concern for researchers.

### 2.3. Stability Studies of PSC

Several groups have attained the promising PCE of perovskite, including novel structures and different material designs. However, despite achieving favourable output, PSC still retains critical barriers regarding device stability the hindering of commercialisation.

However, perovskites are disposed to degradation mechanisms when exposed to ambient atmosphere, elevated temperature, light soaking, UV light, and many other critical factors. Due to all factors, PSC devices cannot achieve market requirements presently [58]. Thus, it is imperative to understand degradation mechanism occurrence in PSC devices and other associated components, e.g., HTL, ETL, and encapsulation, in order to achieve long-term stability of the devices. However, PSCs present improved efficiency in comparison with conventional DSCs, including two first-generation silicon solar cell devices. The stability issue of PSC devices has been considered in various reviews [59]. Still, more attention is required to resolve the degradation mechanism and stability of PSC devices to acquire higher PCE, good reproducibility, and a long lifetime of the PSC system. Some other components, such as the hole transport layer, $TiO_2$ scaffold, and electrodes in the PSC, are also sensitive to temperature, humidity, oxygen, and illumination. A handful of researchers have reported their work to enhance the stability and performance of PSCs [60]. A review of the chemical stability of PSCs against oxygen, moisture, solution processes, UV light, and thermal stress was presented [61]. Afterwards, much advancement in this field was reported. Hongjin et al. overviewed the PSC device decomposition and degradation due to environmental factors and other device components [62].

Wei-Nien Su et al. reviewed the moisture, light, oxygen, and temperature influence on the functioning of perovskite devices that emerge in the degradation mechanism [63]. Factors that can cause device degradation from device components include the perovskite material, device architecture including the $TiO_2$ layer, the inverted structure based on PEDOT: PSS and mesoscopic based on $TiO_2$, the HTL, the metal electrode, and the reaction on the interface. Nam-Gyn et al. emphasised the stability of bulk perovskites and discussed other associated factors, including selective contacts, interface interplay, and correlated intrinsic and extrinsic approaches [64]. The replacement of graphene as an electron transport material and hole transport material and the corresponding strategy to improve stability was also exploited [65]. Nitride-based two-terminal tandem solar cell consisting of $In_xGa_{1-x}N$ as the top cell and a $FAPbI_yBr_{3-y}$ as the bottom cell is optimised through theoretical approach. The power conversion efficiency of 25.17% for the minimum value of the current matching factor of $0.15\,mA/cm^2$ has been achieved. The results indicate the potential for efficient PSC in practical application [66].

The mechanism has been briefly discussed in terms of the stability of the following section of PSC under different factors causing degradation.

### 2.3.1. Crystal Structural Stability

The perovskite material has the general crystal structure of $ABX_3$, as described earlier.

The predicted steady-state performance and stability of standard PV modules is 20 to 25 years; however, depending on various characteristics and properties of the components of PV devices, such stability has not been demonstrated in the last few years.

The structural stability of PSC contemplates the capability of the crystal structure to remain stable against a wide range of factors. The crystal symmetry of halite perovskite can be obtained by maintaining a feasible tolerance factor. That tolerance factor of the crystal structure can determine the rough estimation of the stability and the deformity of the crystal structure. Moreover, it also provides an idea of the phase structure in terms of whether it is cubic or whether it deviates into other shapes such as orthorhombic or tetragonal phase [67].

Methyl-ammonium lead iodide is the most studied and efficient material for the perovskite light-absorbing layer. However, the toxic nature of lead can lead to the production of degradation of PSC devices. Some researchers emphasised replacing lead with other metal ions for further large-scale manufacturing [68]. Several other inorganic cations, organic cations, and halide anions have been integrated into PSC structure engineering to improve the stability and efficiency of the PSC devices [69].

It is also noteworthy that $MAPbI_3$ has reported a tetragonal phase to exist, even after heating at a temperature of 373 K [70]. This indicates the stable nature of the tetragonal

phase in thin films for a specific temperature, but there is an ambiguous phase transition between the tetragonal and cubic phases. Perovskite materials possess different phases depending on the variation of temperature. Perovskites display a stable orthorhombic phase when the temperature is below 100 K. With an increased temperature to 160 K, it displays the tetragonal phase and replaces the original orthorhombic phase [71]. When the temperature increases to 330 K, the cubic phase replaces the tetragonal phase [72].

The phase transition from a tetragonal to a cubic structure can be due to nucleation of lead iodide at the interface due to the intrinsic degradation process. The activation energy related to the dynamic exchange of proton-originating volatile species was found to be $\approx$1.54 eV during inert condition without the involvement of water and decreased to $\approx$0.96 eV with water molecules [73].

The incorporation of organic molecules (FA) and inorganic molecules ($PbI_2$) is another strategy to improve [74] the stability of perovskite against air, the ambient environment, and vacuum. Formamidinium has been reported to obtain phase transition at high temperatures, indicating more stability than $MAPbI_2$. Moreover, another report suggested light soaking as a cause of the reversible phase transition of perovskite materials [75]. However, more research attention needs to reveal the reason behind this behaviour.

The effect of mesoporous $TiO_2$ to retard the phase transition from $\alpha$ to $\delta$ was reported, as shown in Figure 6 [76]. The degradation possibly occurred due to the encapsulation of $\alpha$-$FAPbI_2$ with mesoporous $TiO_2$ and the corresponding synergistic effect and increased energy level for a phase transition.

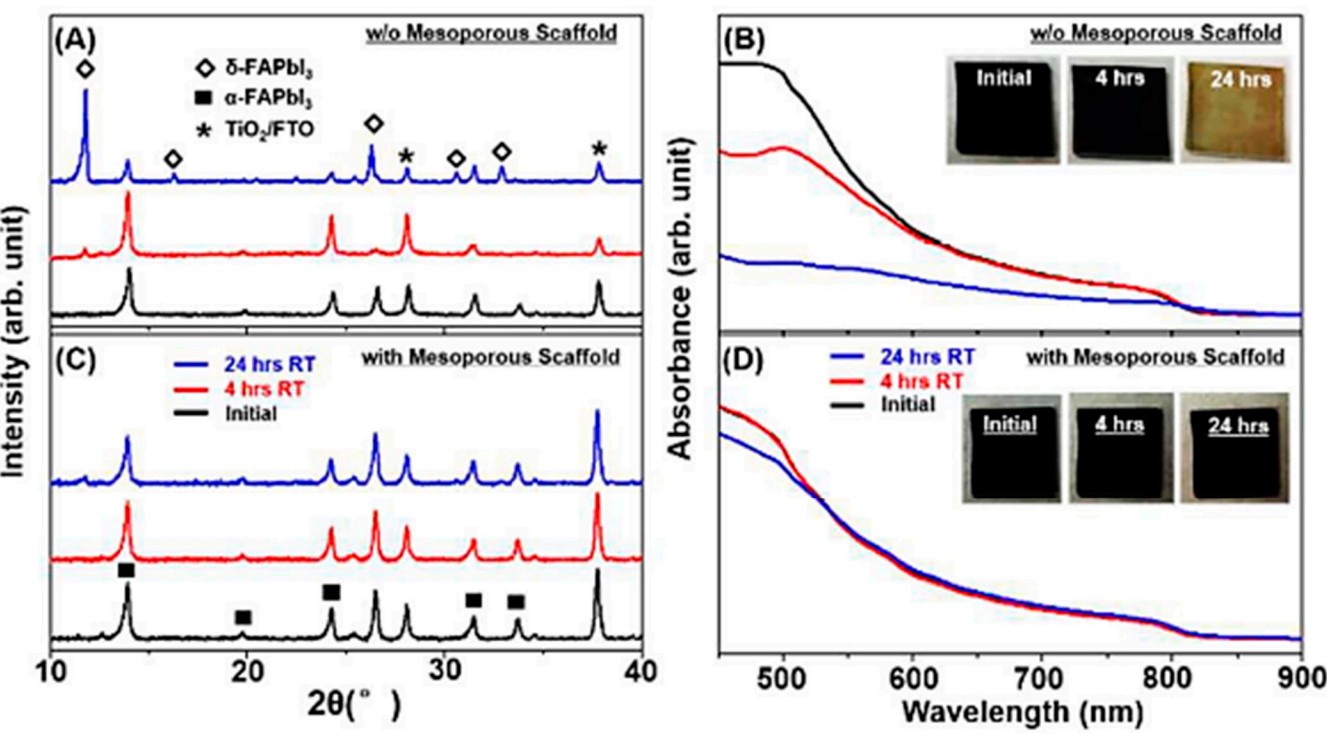

**Figure 6.** Perovskite thin-film XRD pattern and UV-VIS spectra of pure-phase $\alpha$-$FAPbI_3$ before and after exposure to the ambient environment (RH, 45 $\pm$ 5% relative humidity): (**A**,**B**) without mesoporous $TiO_2$ scaffold (planar) and (**C**,**D**) with mesoporous $TiO_2$ scaffold. (Reproduced from [76] with the permission of The Royal Society of Chemistry).

Integration of $CsPbI_2$ has been reported for better stability of the perovskites phase of $FAPbI_2$ [77]. $CsPbI_2$ and $FAPbI_2$ are capable of mixing in the perovskite phase. However, because of the similarity of volume and structure of $FAPbI_2$ and $CsPbI_2$ and the dissolubility of the latter, they show instability and can lead to further massive energy loss [78]. The amalgam of $CsPbI_2$ and $FAPbI_2$ tuned the Goldschmidt effective tolerance

factor and showed improved stability against humidity, resulting in better PSC device performance [77]. CsPbI$_2$Br was studied and presented stabilised PCE of 5.6% with J-V scanning efficiency of up to 9.8% for PCS devices [79].

MAPbI$_3$ showed a lattice phase change near 55 °C. Temperature is for the real solar cell working temperature. If this type of cell is encapsulated from moisture, it can perform at up to 80 °C. If the temperature is limited below that value, heating will not be a problem, even in concentrated photovoltaic systems. This idea has been experimentally performed to show a retained efficiency of 92% of PCE after exposure of 63 h with a 0.2% loss of photocurrent per hour [80].

2.3.2. Effect of Humidity

During assembling and testing the device, oxygen, moisture, humidity, and high-energy photons from UV can directly affect the stability of the perovskite layer. Most organometal halides are highly sensitive to moisture, and the crystallinity of perovskites can be damaged due to the presence of excess water. Oxygen degradation is one of the greater concerns among all factors, as mentioned earlier in a dry atmosphere. Oxygen can bond with perovskite ions such as MA$^-$, Pb$^+$, and I$^-$ and form electronic traps and charge barriers. Severe oxygen degradation can occur on the perovskite film surface due to Pb–O bonds [81]. However, in PSC, moisture degradation can be protected with an encapsulated device and in a controlled operating environment.

However, some authors suggest that the presence of humidity or moisture is beneficial for forming perovskites [82]. Liduo et al. proposed the decomposition process of CH$_3$NH$_3$PbI$_3$ to investigate the humidity factor. They stored films at 35 °C with 60% relative humidity (RH) for 18 h. X-ray diffraction and UV-visible spectrum were used for characterisation before and after storage. The absorption rate and phase transition were measured, and the decomposition process of CH$_3$NH$_3$PbI$_3$ was proposed as follows:

$$CH_3NH_3PbI_3(s) + O_2 \rightleftharpoons 2I(s) + 2H_2O \tag{1}$$

$$CH_3NH_3I(aq) \rightleftharpoons CH_3NH_2(aq) + HI(aq) \tag{2}$$

$$4HI(aq) + O_2 \rightleftharpoons 2I_2(s) + 2H_2O \tag{3}$$

$$2HI(aq) \rightleftharpoons H_2 + I_2(s) \tag{4}$$

Here, perovskite hydrolysed directly into PbI$_2$ in the presence of moisture, and then methylammonium iodide was decomposed to produce HI and later departed in the presence of oxygen, which was due to exposure to UV light. Kelly et al. demonstrated a positive correlation between humidity and PCE using in situ absorption spectroscopy and in situ grazing incident X-ray diffraction. Their work showed improved PCE by carefully controlling the relative humidity by adjusting the constant flow rate of saturated water vapours and dilatant carrier gas particles. Adding the HTM layer was also able to reduce the moisture rate [83]. The absorption of perovskite at room temperature, stored for 14 days in the dark at 50% relative humidity (RH), was reduced in all the visible range of the spectrum [84]. The XRD pattern displayed some diffraction peaks after storage in the dark, which could be indexed as (CH$_3$NH$_3$)$_4$PbI$_6$ 2H$_2$O, but no peak was seen for PbI$_2$. CH$_3$NH$_3$PbI$_3$ was combined to H$_2$O to originate a hydrate product which was similar to (CH$_3$NH$_3$)$_4$PbI$_6$ 2H$_2$O:

$$4CH_3NH_3PbI_3 + H_2O = (CH_3NH_3)_4PbI_62H_2O + 3PbI_3 \tag{5}$$

Therefore, reversible degradation of $CH_3NH_3PbI_3$ can occur when stored in the dark or processed in a vacuum. The absorbance spectra of $CH_3NH_3PbI_3$ and $PbI_2$ were very close, indicating the transformation of $CH_3NH_3PbI_3$ into $PbI_2$ under light.

Heat treatment was another solution to cure the degradation of $MAPbI_2$ perovskite in the presence of humidity [85]. Moreover, some materials showed almost zero PCE of the device under the high temperature of 55 °C in air and for the internal device at 85 °C with a RH of 80%. Due to such severe sensitivity to moisture, the fabrication process must be processed in the glove box filled with inert gas [86,87]. Kwon et al. [88] developed poly [2,5-bis(2-decyldodecyl)pyrrolo[3,4-c]pyrrole-1,4(2H,5H)-dione-€-1,2-di(2,2′-bithiophen-5-yl) ethene](PDPPDBTE), which possesses excellent hydrophobic properties to pervade water to the perovskite layer. The stability test was conducted for Spiro-MeTAD and PDPPDBTE; Spiro-MeTAD showed that 27.6% of PCE decreased after 1000 h of the initial value, whereas the PDPPDBTE-based device showed better performance in terms of stability than Spiro-MeTAD. This experiment was performed at room temperature without encapsulation and with ≈20% RH.

The degradation mechanism of $CH_3NH_3PbI_3$ exhibiting smooth morphology was studied by spectroscopic ellipsometry characterisation. The result showed that degradation might occur due to the elution of $CH_3NH_3I$ and $PbI_2$ formation, as well as the hydration of $CH_3NH_3PbI_3$ by $H_2O$ integration, as shown in Figure 7 [89].

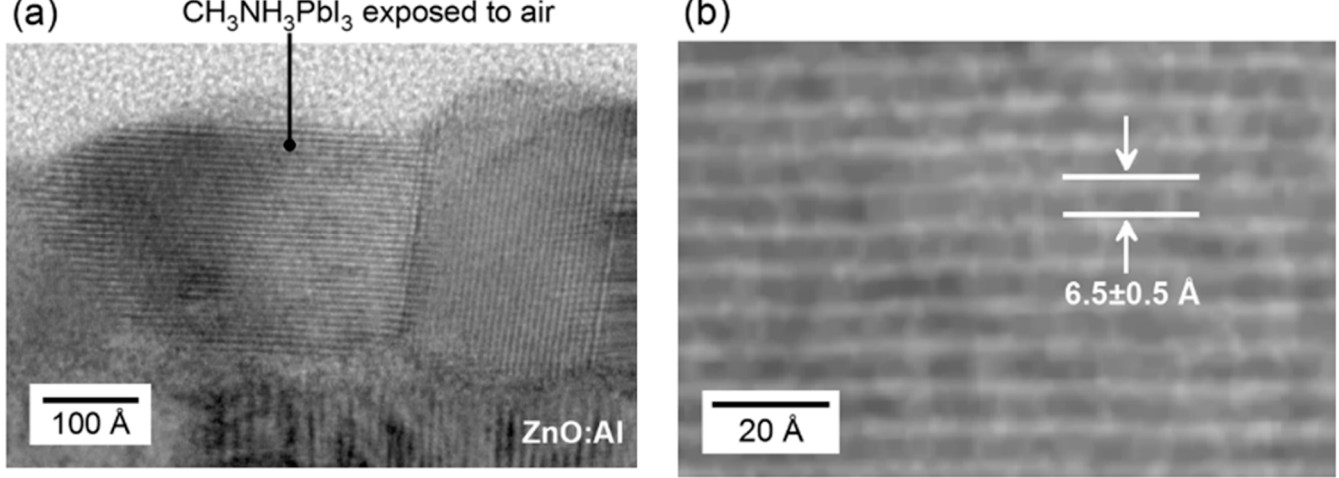

**Figure 7.** (**a**) Cross-sectional TEM image after the air exposure at 40% RH and thermal annealing. (**b**) Enlarged image of the same region. (Reproduced from [89] with the permission of the *Journal of Applied Physics*).

Pristine $CH_3NH_3PbI_3$ layers were investigated by Martin Ledinsky and co-workers. They studied the Raman spectrum and showed the presence of the $PbI_2$ layer as the final degradation product [90]. Moreover, in situ electrical resist measurement and X-ray diffraction analysis were performed to study the interaction between moisture and perovskite film. The results showed that exposure to moisture chemisorption phenomena occurred for a short time, and a decrease in resistance was reversible. However, by extending time, this absorption occurred due to chemical reaction transformation by hydration and consequently the decomposition of perovskite film [91]. An ab initio molecular dynamics simulation was used to investigate the interaction phenomena between the $MAPbI_3$ layer and the present water environment. The results showed that the terminated MAI surfaces went through salvation due to water and Pb interaction, which caused the I atom to release the surface. An intermediate hydrate phase was able to occur due to the incorporation of one water molecule into the terminated $PbI_2$ [92]. A planar perovskite/perovskite tandem solar cell was fabricated using the ALD technique. The quick charge transportation between the interfaces due to ALD AZO's tunnelling effect was considered. A comprehensive study

was carried for the current matching condition. They achieved PCE of $\approx$31% with a realistic $V_{OC}$ of 2.16 V, an FF of $\approx$82.2%, and a $J_{SC}$ of 17.65 mA/cm$^2$. ALD AZO also played a fencing role to intercept oxygen and water from entering the device, resulting in the significantly improved stability of the tandem device [66].

To summarise, the CH$_3$NH$_3$PbI$_3$ compound with a certain amount of water can form an intermediate compound, leading to a reversible process in an inert atmosphere. However, CH$_3$NH$_3$PbI$_3$ can directly decrease to PbI$_2$ in the presence of water and is entirely irreversible. Nevertheless, to prevent voids and holes, the role of perovskite is crucial. A multi-layer encapsulated PSC scheme with the Al2O3 layer and hygroscopic layer deposition was demonstrated. This work exhibited excellent properties to resist water immersion for 5 h with a loss of only 2% efficiency [93].

### 2.3.3. UV Light Stability

Apart from the moisture present in the air, UV light exposure was demonstrated, which strongly affected the performance of PSC devices. A multidimensional PSC system is the best strategy to cope with light behaviour [94]. The degradation mechanism of perovskite by UV light could occur due to integration of TiO$_2$ mesoporous layer in the fabrication of PSC. TiO$_2$, having a 3.2 eV bandgap, is an excellent catalyst for oxidising water to produce oxygen radicals and an oxidising organic compound. Upon UV light exposure, TiO$_2$ can extract an electron from I$^-$ (iodide) when used as a photoanode, such as in typical DSSC, leading to the destruction of the crystal structure of perovskite and building up the strong ionic reaction of organic cations. In this process, extracted electrons trapped by vacant sites will recombine with excess oxygen molecules and generate O$^-_2$ [95]. Upon illumination of UV light, oxygen is removed from the surface where it is absorbed, leaving vacant sites behind, which serve as a trap site for electrons. These trapped electrons are not mobile and may combine with holes in HTM, resulting in a low photo-generated current. Hitoshi et al. analysed CH$_3$NH$_3$PbI$_3$ transformation to PbI$_2$ by decreasing the UV-VIS absorption spectrum and XRD results. They proposed that extracted electrons from I$^-$ by the TiO$_2$ layer degrade the structure stability of the perovskite by originating I$_2$.

As followed by exposure to UV light:

$$2I^- \rightleftharpoons I_2 + 2e^- \text{ (reaction between TiO}_2 \text{ and CH}_3\text{NH}_3\text{PbI}_3) \tag{6}$$

$$3CH_3NH^+_3 \rightleftharpoons 3CH_3NH_2 \uparrow + 3H^+ \tag{7}$$

$$I^- + I_2 + 3H^+ + 2e^- + 3HI \uparrow \tag{8}$$

Followed by the continued elimination of H$^+$ and the evaporation process of CH$_3$NH$_3$PbI$_3$, the reduction of I by the interaction of the TiO$_2$ layer and CH$_3$NH$_3$PbI$_3$ on the interface occurs. The researchers integrated Sb$_2$S$_3$ between TiO$_2$ and CH$_3$NH$_3$PbI$_3$ at the interface to improve the stability of the perovskite upon UV light and noticed a significant enhancement in the stability by using this blocking layer. This worked as a deactivator for the I$^-$/I$_2$ reaction at the TiO$_2$ surface, which enhanced stability.

Inserting of the CsBr layer between the ETL- and CH$_3$NH$_3$PbI$_{3-x}$Cl$_x$-based absorbing layer was proposed to lead to UV light stability with a planar-based structure. They acquired 70% of initial PCE after UV light exposure for 20 min, and the control device under the same operating conditions degraded to zero in the air [96]. Dong et al. reported amino acid as an avert candidate to prevent decomposition of the perovskite layer under illumination by hydroxyl radicals and superoxide anions generated by O$_2$ and H$_2$O present at the interface of the TiO$_2$ layer [97]. Another way to overcome the instability originated by UV light absorption by TiO$_2$ is by applying a U-V filter in front of the TCO substrate before the deposition of the TiO$_2$ layer. However, as compared to other approaches, the U-V filter might lead to unavoidable fabrication costs due to extra material costs, loss in photogenerating a current, and a further decrease in the efficiency rate.

$Al_2O_3$ has shown a promising scaffold as a substitution for the $TiO_2$ layer. The stability of PSC was examined for over 1000 h at 40 °C, encapsulating a window glass lid and epoxy resin in a Ni-filled glove box, whereas 3 h exposure of $TiO_2$-based cells caused degradation to almost zero. Therefore, 5% maintenance of PCE was observed within the first 200 h due to a decrease in F.F and Voc [98]. A comparison of the $TiO_2$ layer $Al_2O_3$ scaffold stability was performed by Flannan T. F. O'Mahony et al. in an ambient environment. They found a rapid degradation mechanism for the $Al_2O_3$ scaffold, which might have occurred due to the parasitic reaction of the oxygen molecule and photogenerated electrons, as shown in Figure 8 [99].

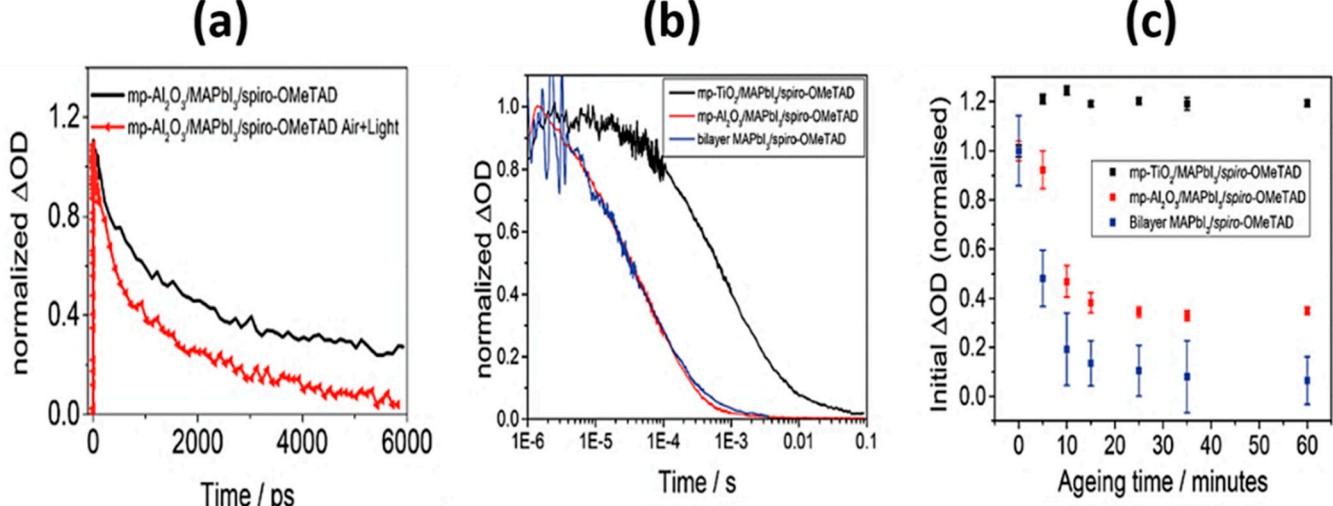

**Figure 8.** (**a**) Time-resolved transient absorption at 780 nm for the device with the configuration of mp-$Al_2O_3$/MAPbI$_3$/Spiro-OMeTAD (black) exposed to air and ambient at a 1 ps time-delay to highlight differences in recombination kinetics. With light MAPbI$_3$/Spiro-OMeTAD (red) films. (**b**) Time-resolved transient absorption at 1600 nm for pristine mp-$TiO_2$/MAPbI$_3$/Spiro-OMeTAD (black), mp-$Al_2O_3$/MAPbI$_3$/Spiro-OMeTAD (red), and bilayer MAPbI$_3$/Spiro-OMeTAD (blue) films. (**c**) Relative yield of permanent charge separation as a function of ageing time in atmospheric conditions under ambient light. (Reference [99] published by The Royal Society of Chemistry).

The photogenerated electrons might migrate to combine oxygen molecules, leading to the formation of superoxides. Then, these superoxides cause the degradation of perovskite by the deprotonating of methylammonium cation [100]. However, avoiding parasitic reactions between photogenerated electrons and oxygen present in the environment can improve the stability of PSC. The fast electron extraction from the perovskite layer with meso-$TiO_2$ as an acceptor can help to circumvent parasitic reactions. ETM as a substitution for $TiO_2$ can be helpful in improving the stability of PSCs [101]. $SnO_2$-based PSC performed well to remain stable for 700 h of storage as compared to $TiO_2$-based PSC [102]. A highly stable $SnO_2$ nanocrystal layer was used to show 90% retained PCE of its initial value of 18.8% after storage of 30 days with <70% RH at an ambient environment [103]. PSC performance under prolonged light exposure is not only due to high-intensity light or optoelectronic effects. It can behave worse on the mechanical integrity of the device by inducing high temperatures [104]. A highly efficient $Mg_xZn_{1-x}O$-based (MZO-based) PSC was reported. MZO has a high conduction band and electron mobility when compared with traditional electrode $TiO_2$. These properties can reduce charge accumulation at the MZO/perovskite interface and increase the charge transfer between the interfaces. The encapsulated device retained 76% of the initial Jsc after one year of aging and 8 h UV irradiation, while for $TiO_2$, only 12% of its initial Jsc was retained. The work outcomes show great potential for MZO ETL in perovskite application [66].

2.3.4. Thermal Stability

Like moisture, the temperature can cause the degradation of perovskite material, and other main defective components can be a hole transition material. According to international standards, 85 °C is the minimum temperature required to be the best competitor to other solar cell technologies [105]. However, organometal halide perovskite materials have been reported to maintain the stability above 300 °C reported earlier. The degradation of the perovskite material was reported to operate below a 140 °C temperature in the literature [106]. Typically, perovskite material fabricated by solution process requires an annealing process, where 80 °C is the minimum temperature for the complete decomposition of $PbI_2$ and $CH_3NH_3I$ in $CH_3NH_3PbI_3$. $CH_3NH_3PbI_3$(MAPbI$_3$) has been widely discussed as generating thermodynamic degradation due to the origin of volatile molecular defects [107]. Perovskite material using $CH_3NH_3PbI_{3-x}Cl_x$ was fabricated in a $N'_2$ atmosphere at up to 100 °C temperature, and a 3D perovskite structure was formed at 90 °C, whereas the degradation mechanism was seen at 100 °C [108].

Fan et al. studied the crystal structure of the MAPbI$_3$ microplate using a high in situ resolution transmitted electron microscope. They showed that after 100 s of heating at 85 °C, almost 75% of the transformation of the initial tetragonal phase to trigonal $PBI_2$ occurred [109]. The degradation mechanism occurred due to breaking the weak Pb-I-Pb bond along the (001) direction. This work also revealed that MAPbI$_3$ hydration does not originate during the degradation mechanism [83], as they conducted their experiment in inert gases and dry environments.

Han et al. investigated the thermal stability of the PSC device using a temperature-controlled environmental chamber operating at −20 °C to 200 °C [110]. The environmental temperature was lower than the actual cell temperature, e.g., 55 °C (85 °C). Cross-sectional focused ion beam scanning electron microscope (FIB-SEM) analysis was performed to study the degradation mechanism of $CH_3NH_3PbI_3$ for 500 h. Different components of the encapsulated device were observed to be defective due to direct exposure to one sun illumination. The degradation of silver layer formation of voids in HTM and the light-absorbing perovskite layer, the separation of the perovskite layer and $TiO_2$ layer, and the main formation of $PbI_2$ were observed. The degradation possibly occurred due to an encapsulated device's reaction between Ag and HI gas. Therefore, the replacement of the Ag layer by appropriate metal and highly heat-repellent material and encapsulation is recommended by the authors. Herz et al. observed MAPbI$_3$ at an operating temperature of 100 °C and showed a gradual shift in the bandgap using PL and transmittance measurements. Their results suggested that inherent thermal degradation of the PSC devices operating at low temperatures can limit its use at the commercial level, as shown in Figure 9 [20].

Wu et al. reported improvement in crystallisation and oxidation in Spiro-MeTAD-based perovskite devices after annealing. This was better for hole transport and transfer. Consequently, higher $I_{sc}$ was achieved. Therefore, due to Li-TFSI transfer to the $TiO_2$ layer and evaporation occurrence of 4-tert-butylpyridine, the fermi level of $TiO_2$ shifted down. Consequently, low FF, reduced PCE, and $V_{oc}$ were acquired. However, the absence of HTM is a suitable choice for a capable PSC device to be stable. A PCE of 10.5% was obtained without HTM layer integration in the perovskite structure [37]. Li et al. tested PSC devices fabricated with solution processing with meso/$TiO_2$/$ZrO_2$ scaffold coating and back contact of carbon black, being encapsulated to avoid moisture. They showed the same PCE after exposure for 7 days in an actual temperature and UV condition, as shown in Figure 10 [58].

Han et al. also observed HTL-free PSC devices with carbon as the back contact without any ceiling, and after exposure to the sun for 1008 h in the ambient environment, they acquired a greater level of PCE than the initial amount [111].

The performance of PSC devices at a prolonged light period was theoretically studied. In a temperature range of 300–360 K, different PV parameters such as Voc, PCE, and Jsc were analysed.

A declined nature was seen for Voc and PCE with a temperature rise above room temperature. The experimental results for this work exhibited the decaying nature of Jsc vs. temperature plot shown in Figure 11. Alternatively, the effect of temperature on the PCE of PSC was more negligible at a higher illumination than silicon solar cells. Recently, zinc(II) bis(trifluoromethanesulfonyl)imide (Zn(TFSI)$_2$) was proposed as a potential candidate to replace LiTFSI. The thermal stability of PSCs was investigated using Zn(TFSI)$_2$ as a dopant for Spiro-MeOTAD. As a result, the morphological stability of the Spiro-MeOTAD film with a higher Tg was observed. Instead of inefficient hole extraction and decline, Rrec, as in LiTFSI-based Spiro-MeOTAD Zn(TFSI)$_2$, maintained photovoltaic performance, despite the development of pernicious pinholes [66].

A highly efficient device with a PCE of 23.36% and improved thermal stability over 1400 h operation under 80 °C was presented by employing benzamidine hydrochloride as a spacer [66].

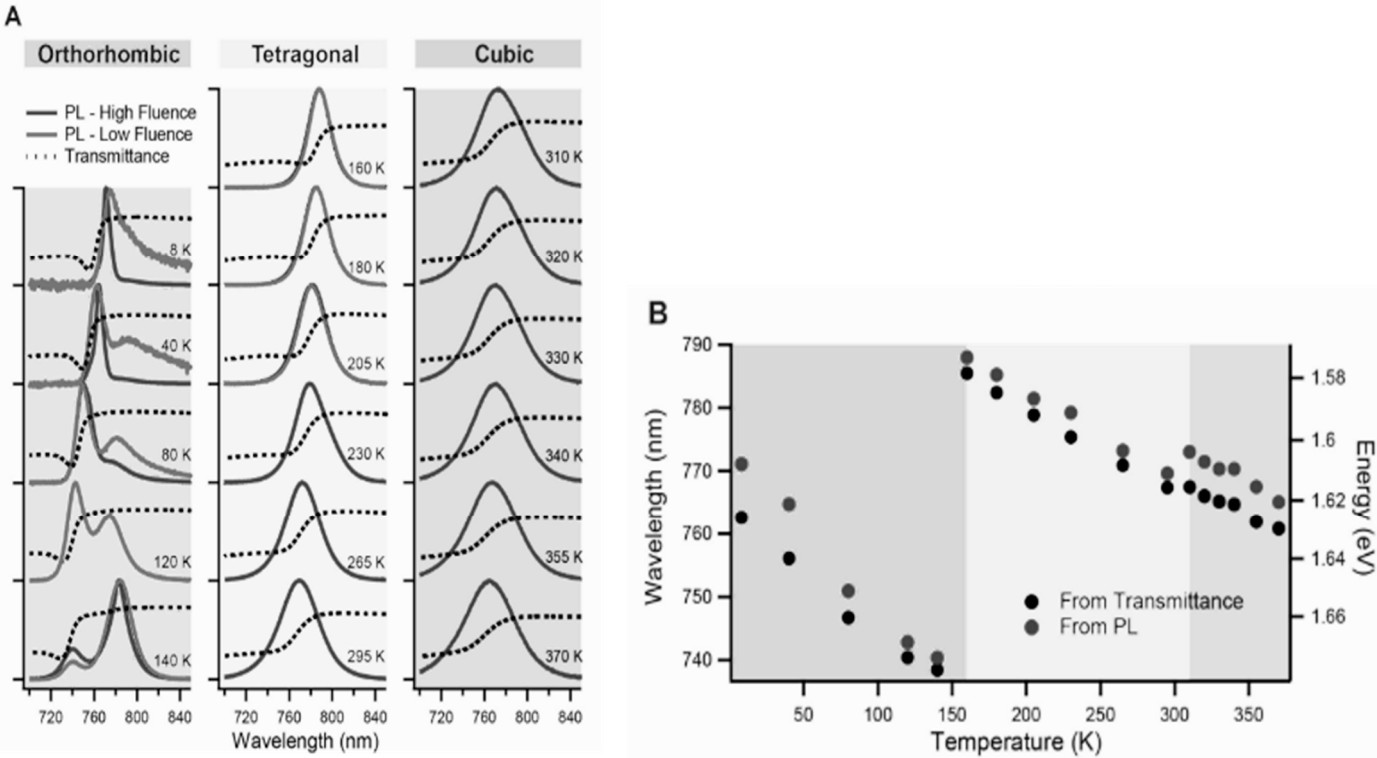

**Figure 9.** (**A**) Transmittance (as dashed lines) and normalised steady-state PL (as solid lines) spectra of MAPbI$_3$ measured at temperatures ranging from 8 k to 370 K. Black ticks on the y-axis indicate values of 0 and 1 below and above the respective curves. PL for high fluences ranging from 45 to 120 μJ cm$^{-2}$ (red lines) and low fluences ranging from 0.20 to 10 μJ cm$^{-2}$ (blue lines) are included. The shading corresponds to three different phases as labelled at the top of the figure. (**B**) PL peak wavelength (energy) and absorption band-edge wavelength (energy) as a function of temperature. The same shading scheme is used as in (**A**). (Reproduced from [20] with the permission of Advanced Functional Materials).

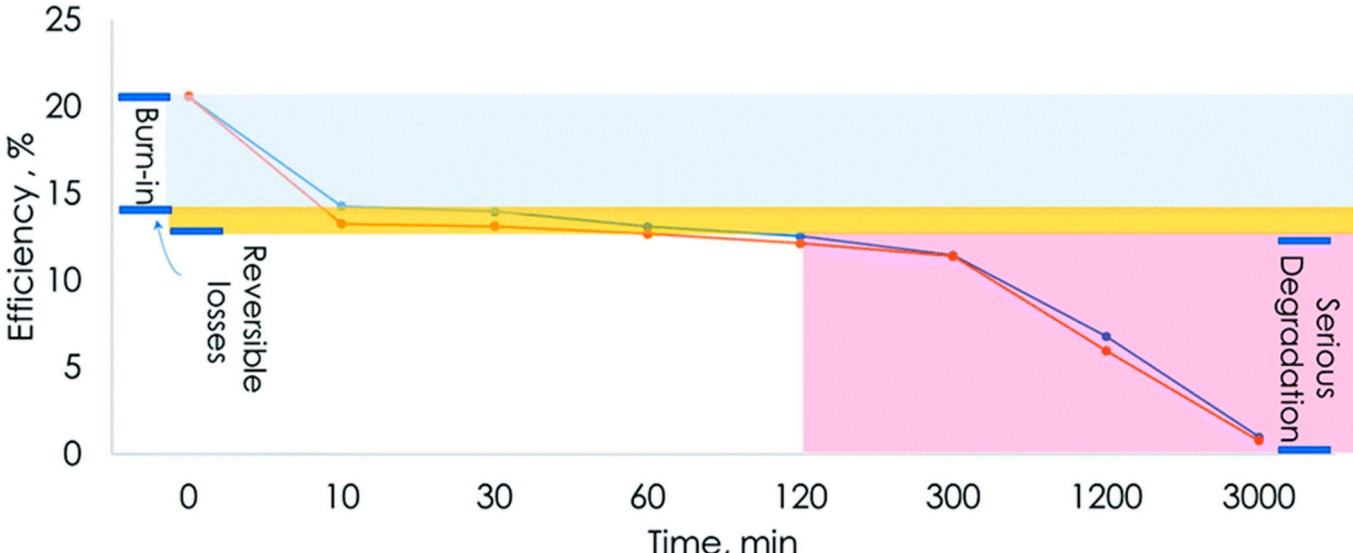

**Figure 10.** (**a**) J-V curves for the device under simulated standard AM1.5 solar irradiation measured at room temperature. (**b**) Indoor heat stress test of PSCs. (**c**) Evolution of relative device performance parameters for PSCs aged under Ar at 45 °C and at maximum power point tracking conditions. (Reproduced from [58] with the permission of ENERGY TECHNOLOGY).

**Figure 11.** Decline curve of efficiency over time under a 1.78 sun.

### 2.4. Hysteresis

Another challenging characteristic of PV solar cells to restrict further development is the presence of I-V hysteresis when measured by applying different voltage sweep rates in both forward and reverse directions [112]. Usually, the enhanced performance is acquired with a forwarding bias, followed by a backward bias condition. The two significant categories of hysteresis are typical and inert hysteresis, which can be found together or separately, depending on the applied pre-pole bias [113]. The possible factors causing the mechanism of the hysteresis can be the capacitive effect, trapping of charge carriers at the interface of the perovskite, ionic replacement, and ferroelectric polarisation [114–117]. Several reviews have proposed different hypotheses about these factors [118]. Moreover, the origin of these factors, intrinsic or extrinsic, is still unrevealed [119,120]. The solution to the hysteresis issue is essential for successfully developing PSC device characteristics, improving stability, and enhancing breakthrough advancement in associated applications [121,122]. Recently, some reports claimed to acquire both PCE and device stability with low I-V hysteresis during the fabrication process of the PSC device. However, I-V hysteresis is still under consideration to identify the origin and other related mechanisms governing the hysteresis mechanism during PSC characterisation [123,124]. Van Reenen et al. proposed theoretical studies to report the combined effect of ion replacement and trapping of charge carriers causing hysteresis in PSC devices. They observed that for the reduction in the hysteresis effect and enhanced performance of PSC, the ionic migration and grabbed site recombination should be limited [125].

Their work also suggested that high-density charges offer unfavourable biasing by allowing non-radiative trapped charge recombination, which results in less photocurrent and low PCE of the device. On the contrary, charge with low-density offers favourable biasing by allowing the amalgam of trapped free holes and electrons, leading to better performance of the PSC device. Kutes et al. studied the first observation of the presence of a ferroelectric domain ($\approx$100 nm) in terms of the improved quality of $\beta$-$CH_3NH_3PbI_3$ thin film of perovskite. They observed that poly with DC bias reverse switching of the ferroelectric domain could occur [126].

This phenomenon was observed by Dualeh et al. and presented the occurrence of electron conduction occupying ionic migration [127]. The production of an acceptor or shallow donor for $PbI^{2+}$, $I^-$, and $CH_3NH_3^+(MA^+)$ vacancies was also suggested [128,129]. Moreover, ionic disorder over electronic disorder in $CH_3NH_3Pb_3$ is analogous to cation and anion vacancies. Ion migration in PSC might cause a stoichiometry change at the vacant contacts leading to the more complex behaviour of the device. However, incorporating methylammonium and iodide ions can help provide doping regions at contacts to influence contact selection, photocurrent properties, and I-V hysteresis. Sainth et al. proposed that cell architecture is likely to influence the I-V characteristics of the cell dependence on selective contacts [28]. They also observed that increased hysteresis by slowing down the steady state was obtained at any applied bias condition without depending on history. They obtained better results of reasonable charge-selected contacts at the interface with forwarding bias conditions. However, the short circuit condition device showed poor performance because of the empty trap site caused by direct charge transfer to adjacent contacts. This state remained until the trap was occupied again.

The capacity analysis is another crucial parameter to detect the cause of underlying I-V hysteresis. These analyses address the kinetic of the charging process and the nature of the charge distribution and solar cell current distribution phenomena. The capacity characteristics of PSC devices have been reported in some reports [114,130,131]. Further, a $CH_3NH_3PbI_3$-based system was proposed to exhibit a significant dielectric constant and a high polarisation factor raised by illumination and extended voltage. Moreover, low-frequency capacitance slows the dynamic process in PSC-originating I-V hysteresis [132]. Various reports revealed that excess capacitance in PSC could be caused by electronic traps occupied in methylammonium iodide films following a given state density [133,134]. There are several other parameters on which I-V hysteresis depends, e.g., the scan rate, interface

of material, illumination condition, biasing history, device structure, reversible light and voltage, and bias pre-conditioning.

Various approaches are discussed in the literature to cope with instability issues and other factors causing low PCE (Table 1). The C-Si/perovskite tandem is one of the emerging ideas to improve PCE by combining two absorbing materials.

**Table 1.** Different strategies to improve output of PSC devices.

| Perovskite Material | Perovskite Composition | Efficiency | Year | References |
|---|---|---|---|---|
| $CH_3NH_3$ mixed | (PEAI) on HC $(NH_2)_2$–$CH_3NH_3$ mixed | 23.32% | 2019 | [135] |
| | | 25.5% | 2021 | [136] |
| **Additive** | | | | |
| Perovskite Material | Additive | Efficiency | Year | References |
| $MAPbI_3$ | Pyrrole | 20.07% | 2019 | [137] |
| $CsPbI_3$ | Bis(pentafluorophenyl)zinc $[Zn(C_6F_5)_2]$ | 19% | 2020 | [138] |
| $Cs_{0.05}(MA_{0.12}FA_{0.88})_{0.95}Pb\,(I_{0.88}Br_{0.12})_3$ | 6-Aminoquinoline monohydrochloride (AQCl) | 21.66% | 2021 | [139] |
| **Defect passivation implement** | | | | |
| Passivation Material | Perovskite Absorbing Layer | Efficiency | Year | References |
| Phenethylammonium iodide (PEAI) | $(FAPbI_3)_{1-x}(MAPbBr_3)$ | 23.32% | 2019 | [135] |
| 4-Tert-butyl benzylammonium iodide(tBBAI) | $Cs_{0.05}FA_{0.85}MA_{0.10}Pb(I_{0.97}Br_{0.03})_3$ | 23.50% | 2020 | [140] |
| Cyclohexylammonium chloride (CYCl) | $FAPbI_3$ | 23.34% | 2021 | [141] |
| **Surface modification** | | | | |
| Device architecture | Fabrication strategy | Efficiency | Year | References |
| $ITO/SnO_2/perovskite/PTAA/Metal$ | Self-assembled facile strategy | 20.30 | 2019 | [142] |
| $ITO/NiO_x/PTAA/(MAPbI_3)_{0.95}$ $(MAPbBr_2Cl)_{0.05}/PCBM/BCP/Ag$ | Spin coating | 21.56 | 2020 | [143] |
| $FTO/TiO_2/perovskite/(Me-PDA)Pb_2I_6/Spiro-OMETAD/Au$ | Perovskitoid surface engineering | 22.0 | 2021 | [144] |
| $FTO/TiO_2/CsPbI_2Br$ [PEVIM]Cl modified/Spiro-OMETAD/Ag | Surface modification | 14.19 | 2021 | [145] |
| $ITO/Cs_{0.05}(FA_{0.92}MA_{0.08})_{0.95}Pb$ $(I_{0.92}Br_{0.08})_3/ETL/BCP/Cu$ | Surface modification | 22.0 | 2022 | [66] |

| **CPV-based PSC** | | | | | | | | |
|---|---|---|---|---|---|---|---|---|
| Perovskite Material | Strategy | Device Area | Lifetime | Efficiency | No. of Suns | Stability | Year | References |
| $FA_{0.83}Cs_{0.17}PbI_{2.7}Br_{0.3}$ | Concentrated light | 9.19 mm$^2$ | 150 h | 23.6% | 14 | 90% of initial η at 10 suns | 2018 | [11] |
| | Concentrated light | | | 26.5% | Theoretical | | 2018 | [146] |
| | Concentrated light | | | 24.93% | Theoretical | | 2019 | [147] |
| $(FAPbI_3)_{0.875}$ $(MAPbBr_3)_{0.125}(CsPbI_3)_{0.1}$ | Concentrated light | 9 mm$^2$ | 5 h | 21.6% | 1.78 | 19% of initial η at 1.78 suns | 2020 | [148] |

## 3. Pathways to Improve Stability through Concentrated Perovskite Solar Cells

Plenty of strategies have been reported to enhance the power output of photovoltaic systems such as mobile ion concentration by considering the electrostatic aspect of migration and accumulation of mobile vacancies at the contacts. However, no considerable results are evident in this case. The experiment's conclusion is consistent with addition of halide such as Pa iodide or Cd, leading to lower number of mobile ions. Evidently, it results in stable devices with better efficiency due to no effect of ion concentration on the device [149].

The concept of a tandem solar cell is the implementation of different absorbers with high absorbance coefficients. For tandem solar cells (TSC), crystalline silicon solar cells are the ideal choice in terms of offering low-bandgap bottom cells due to their appropriate bandgap of 1.1 eV, high $V_{oc}$ of nearly up to 750 mV, and also cost-effectiveness for manufacturing. Therefore, finding a suitable wideband partner has been difficult for researchers. Organic–inorganic metal halide PSCs propose several advantages that make them highly captivating to use as the top cell in silicon-based tandem solar cells. The function of the top cell in TSC is to absorb the incident spectrum with a large bandgap. However, it converts only the higher energy photon at a higher voltage and permits lower energy photons to pass through to be absorbed in the low bandgap bottom cell. In this way, a broad range of the incident spectrum can be harvested to raise the combined obtainable efficiency of the TSC to make it an exciting candidate in the PV era.

While significant challenges exist, perovskite solar cells are still touted as the PV technology of the future, and much development work and research are put into making this a reality. Scientists and companies are working towards increasing efficiency and stability, prolonging lifetime, and replacing toxic materials with safer ones. Researchers are also looking at the benefits of combining perovskites with other technologies, such as silicon, to create what is referred to as "tandem cells". The promise of high conversion efficiency from tandem cells has prompted research efforts to surmount problems inherent in their design and fabrication. Designing tandem cells is a remarkably complex exercise. Different strategies to improve the PCE of tandem PSC are shown in Figure 12.

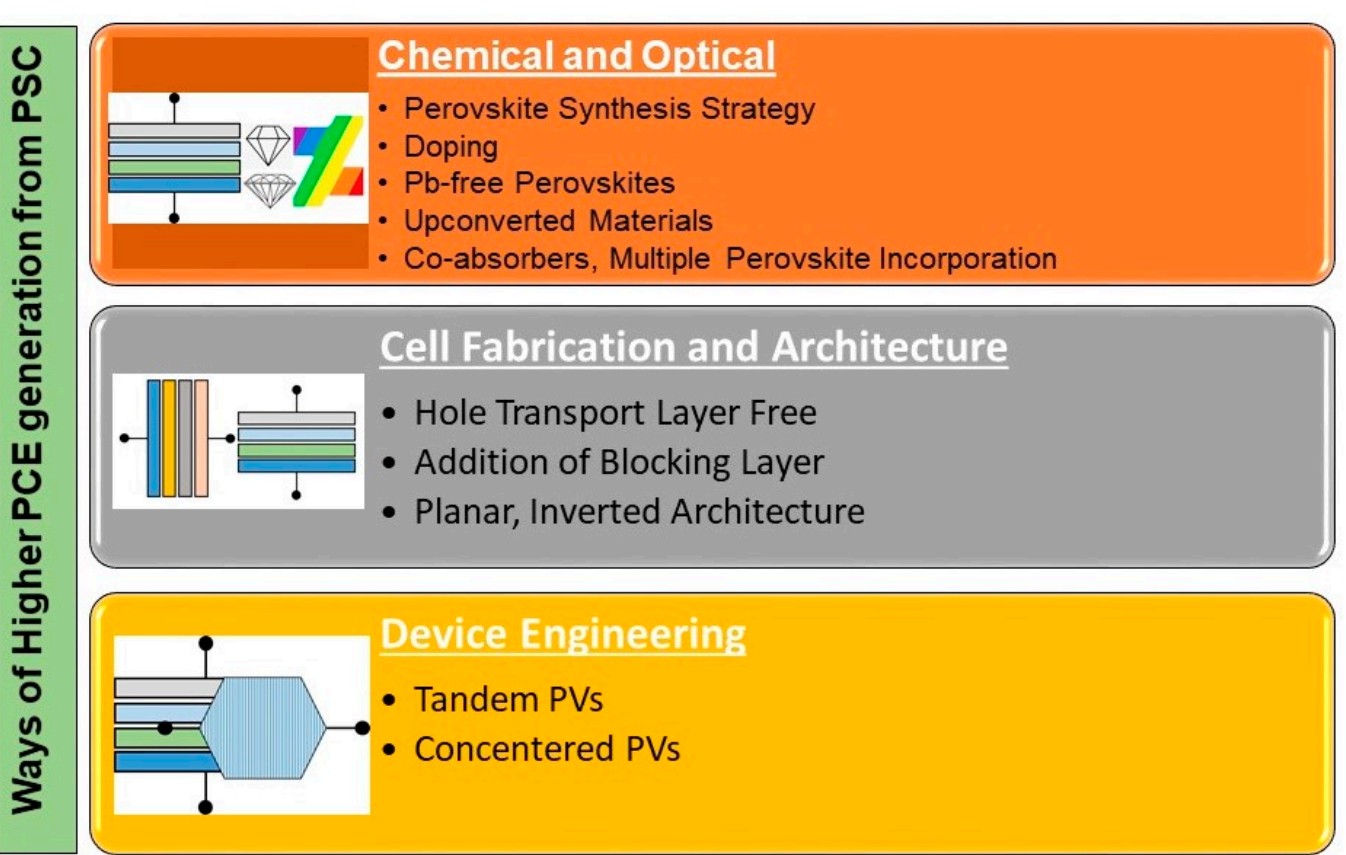

**Figure 12.** Various ways to improve the efficiency of a PSC.

However, perovskite/Si tandem solar cell architecture is reported to achieve an efficiency of over 26% [150,151]. Perovskite materials are cheap and have fascinating optoelectronic properties for integration as the solar absorber, lower recombination defect chances at the interface, ease of fabrication, and a wide choice of fabrication approaches. Thus, perovskites can be used as promising candidates exhibiting distinct properties, as

mentioned earlier, to fulfil the requirement of being a perfect material for tandem solar cells. In all these respects, perovskite-based tandem solar cells can achieve the highest efficiencies [152–154]. Perovskite/Si tandem solar cells have been under consideration, owing to their high demand at the commercial level and astonishing strides with exceptional efficiency of 28% in a few years of development. The single-junction solar cells have already reported acquiring over 26% efficiency, and thus the perovskite/Si tandem will have to achieve over 30% efficiency to compete. In tandem solar cells, the top cell comprising the high bandgap trap, high energetic photons generate a photocurrent at high voltage from the short-wavelength part of the solar spectrum with significant losses. Longer wavelength photons or low-energy photons beyond the bandgap of the top cell are mostly absorbed by the bottom cell as it has a narrow bandgap. Typically, Si is the best option for the bottom sub-cell as it provides better overall performance and optical absorption for near-infrared photons. The important distinction for tandem solar cells is based on their fabrication approaches: (i) single package or two terminals (2-T), (ii) monolithic design, and (iii) mechanically stacked separate call sub-cells that are optically coupled (4-T). In four-terminal tandem solar cells, the top and the bottom cells are fabricated separately and are mechanically placed on one another. Disarrangement allows for the maximisation of the efficiency of the tandem, but the integration cost becomes high [155]. However, in 4-T tandem solar cells, no electrical coupling is involved between the sub-cells, which require only the top cell to be semi-transparent for optical coupling, making them easy to prototype. This is likely the reason explaining why 4-T tandem solar cells were able to achieve a high efficiency of 26.4% with single-junction solar cells [156]. On the other hand, 2-T tandem solar cell functioning is more complicated due to its compatibility between every processing step within all interfaces and layers. The 2-T tandem is fabricated on top of one another but requires careful engineering of its component to achieve higher efficiency. For utilisation of the full theoretical efficiency, for the potential of 2-T tandem solar cells, the sub-cells being integrated with the architecture is required for them to connect properly in series with good functioning recombination layers. The function of the recombination layer is to make the electrical connection between both subshells in the series. It provides good selectivity for charge carriers and favourable energy levels for additional photovoltage and low series resistance without any losses, as well as high transparency for bottom cells.

The Shockley–Queisser limit of PSC could be reached with a 200 nm thickness by integrating a wavelength-dependent angular-restriction design with a textured light-trapping structure. Recently, Nishigaki et al. reported a simulations study based on light absorption characteristics of chalcogenide-based perovskites, indicating a maximum potential PCE of 38.7% in the perovskite/Si tandem structure [157]. Despite the intrinsically excellent optoelectronic properties of organic–inorganic lead halide perovskite, a gap between the theoretical efficiency and the experimental one opens up an ample scope for further investigation and detailed scrutiny to unveil the hidden information of achieving high efficiency and stable PSC.

An approach similar to conventional modelling of the PN junction was reported to analyse the impact of mobile vacancy concentration on the PV solar cell for the recombination process. The work reported minimising the surface recombination at the HTL, yielding large efficiency increments, and then minimising surface recombination at the electron transport layer [149].

An appealing route to unleash the potential of PSC devices at the commercial level was studied using the thermal evaporation fabrication method. The superstarter configuration method is not viable in tandem PSC for stability and the lifetime of the devices due to the deposition of narrow bandgap subcells at the end. The narrowband subcells are easily exposed to air, which makes them unfavourable. A reverse method of deposition was implemented to cut back the oxidation of narrowband gap PSC in the device configuration. An efficiency of 25.3% was achieved for substrate configured device. However for efficient flexibility, all PSC 24.1% and 20.3% efficiency was achieved on copper-coated polyethylene naphthalane and coper metal foil, respectively [158]. Many researchers combined

different materials with different energy gaps using tandem technology. In the traditional evaporation method, several limitations, such as the growth rate of crystallisation, are challenging in terms of the commercialisation of the technique. Recently, handmade sandwich evaporation chamber technology was adopted to solve the issue faced by the traditional evaporation method, and efficiency of 14.8% was obtained for the formed $MAPbI_3$ and 16.25% for $MAPbI_xCl_{3-x}$. The crystanility enhancement and good optical results depict the potential of the work by using an appropriate bandgap of PSC matched with other silicon solar cells to absorb the maximum light spectra to achieve higher output [159].

A carbon-electrode-based PSC device with a sodium azide ($NaN_3$) additive was employed. The non-radioactive charge recombination reduction through scaling down trap density was evident. The champion efficiency achieved with the PSC device was 14.90%. A higher shelf and ambient stability were observed for the additive-based device compared with unmodified devices. Experimental findings suggest the potential of $NaN_3$ additive to PSC for tables and more efficient devices [160]. Recently, another additive engineering approach was adopted to optimise the PV performance by adding the NaF additive into perpovskite film. The additive NaF showed an efficiency of 11.26%, with improved stability, light absorption, and hydrophobicity. The device retained 95% of its initial efficiency after exposure to a humid atmosphere for 2400 h [161].

An experimental investigation was adopted to show the results of the initial degradation of perovskite film under concentrated light. The formation of traps and non-radiative recombination was indicated due to the fall in intensity and current. PSC retained 80% of the initial efficiency after 5 h of continuous illumination. Moreover, the shelf time of the cells in the glove box was about 60 days [162].

An innovative idea of implementing concentrated optics PSC intensified incident illumination and increased accumulative transformation competence. Concentrated photovoltaics (CPV) refers to increases in the solar flux into an absorber using an optical device to reduce the solar cell area, providing higher electrical power. Such a device can be in different categories such as low concentrating, medium concentrating, high concentrating, and ultra-high concentrating photovoltaics. Moreover, CPV also depends on the other aspect of CPV technologies, which includes geometric profiles, tracking methods utilised in the system, types of primary optical devices, and types of secondary optical devices integrated into the system. While high and ultra-high CPV systems achieved the highest recorded efficiency, these systems are unsuitable for new solar cells such as PSC. Therefore, this work is aligned with low-concentrating photovoltaic (LCPV) applications for integrating PSC solar cells.

According to the detailed balance theory, the material bandgap Eg $\sim$ 1.2–1.4 eV is required to recognise highly efficient single junction PSC. The concentrated counterpart, the optimal Eg, has a slightly lower value. Fortunately, the perovskite absorbing layer materials, such as the mixture of Sn-Pb, perfectly meet the requirements of Eg, leading to the high output of PSC. A downside of mixed Sn-Pb is its low performance and poor stability, originating from rapid crystallisation and ease of oxidising.

Some efforts have been made to optimise Sn-Pb mixtures, such as the passivation of layers and composition engineering. Various photovoltaic devices based on $MAPbI_3$ GaAs or other materials have been reported to integrate with CPV technologies.

Recently, a highly appealing alternative to achieving a high PCE of perovskite under concentrated solar light was presented [163]. A PSC with 0.81 $mm^2$ sizes was used with low concentrated PV, and 23.1% efficiency was achieved to open a new area of research in PV technology [11]. Perovskite materials exhibit unique properties to sustain linearity in photocurrent generation, even with an intensity of up to too many tens of suns [80]. Another critical feature of perovskite is its inherent stability under highly intense light, whereas light degradation can occur in other materials with this condition. However, the first new measurements for concentrated perovskite solar cells showed less stability against even one sun [105].

Motivated by these factors, the surprising concept of using Si/perovskite tandem solar cells under concentrated light to enhance PCE was proposed in order to bring about exciting results. This new idea of utilising S/perovskite tendon solar cells as the application of LCPV can bring down the cost of the PV system and enhance performance to move beyond the Shockley–Queisser limit. A real attempt to integrate concentrating optics with the PSC solar cell, where the IV curves of the system are indicated. However, the paper reported that the overall efficiency improvement of 21.6% was achieved with a much larger device size of 9 mm$^2$.

This appears to be the most prominent device reported for integrating CPV optics with the PSC cell. PSC tandem devices with 2T and 4T architecture and tandem PSC with optical concentration for more improvement in device efficiency as upcoming foresight are shown in Figure 13.

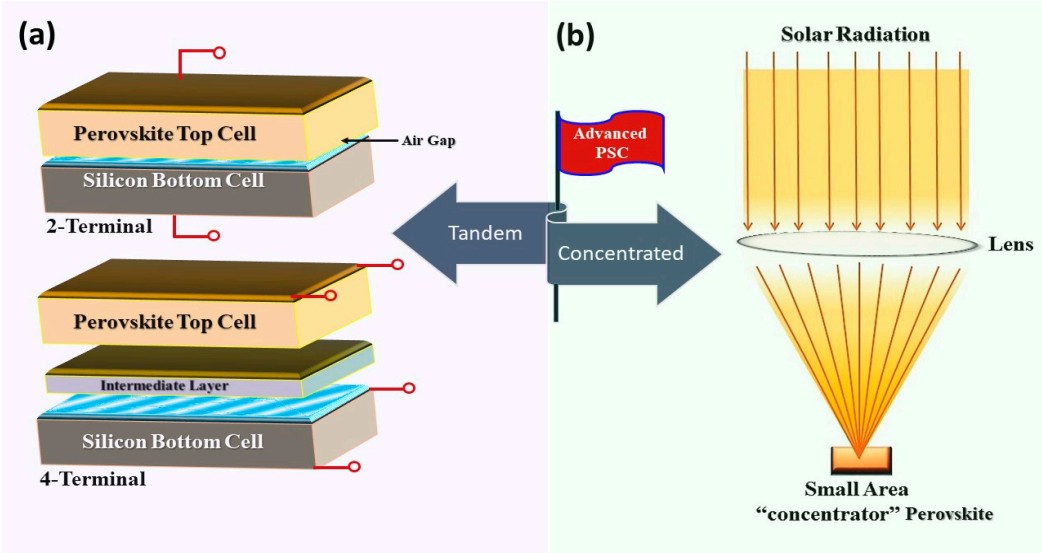

**Figure 13.** (**a**) Schematic diagram of 2T and 4T silicon/perovskite tandem solar cell. (**b**) Single-junction PSC with a small area concentrator as the future direction.

Regardless of extensive improvement in the field, the performance is still far from expectations. A detailed exploration of underlying physics is needed for highly performed PSC, which is a great challenge due to the complications of multi-physics processing.

## 4. Cost-Effectiveness

The cost of PSC is another challenge for researchers. However, the cost of PSC cannot be determined at this stage due to its unavailability at the commercial level. The cost of PSC is dependent mainly on HTM and the back electrode. Scientists must explore less costly materials to fabricate BSc without compromising efficiency. All these routes can lead to deteriorating PCE.

Some strategies such as HTM-ETM free layers, cost-effective black electrodes, carbon-based solar cells, avoiding the glove box, use of inert gas, and low-cost deposition processes can be implemented to cope with the problem. Moreover, the upscaling of PSC from the laboratory to the commercial level and large-area systems can cause a decline in output. A graphical representation of output of small area cell, large area cell and mini-module cell is shown in Figure 14.

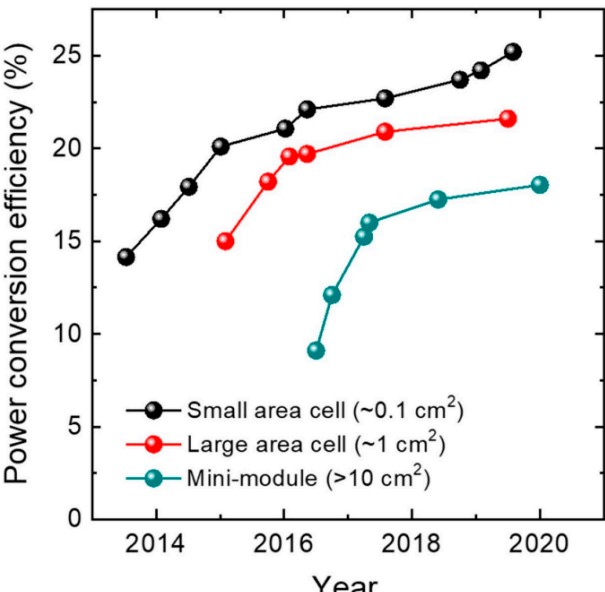

**Figure 14.** PCE improvement of PSCs at different active device areas.

Solution method processing can also be an effective method to lower the cost. However, lifetime and stability are still a concern in boosting the field. The minimum sustainable price value of PSC systems varies between 0.21 USD/W to 0.42 USD/W, which is dependent on the fabrication route and the PSC of the system [164]. However, the PSc with optics cost is still cheaper than its other competitors, such as silicon-based solar cells. Silicon-based solar cells at 0.49 USD/W are not mature technology in the context of technological improvements. Silicon-based solar cells are primarily based on their commercial ground. The cost is mainly dependent on the concentration level. It has been experimentally proved that PSCs are more suitable for >100 concentrations, which is much cheaper than the ≈1000 concentrations used in previous systems. Therefore, the efficiency of PSC is now here as the efficiency of silicon solar cells, and the cost of PSC is much cheaper. The considerable efficiency result in both contexts makes it the most fruitful in the PV era.

## 5. Conclusions

This review article presents a detailed amount of discussion on the gradual progress in perovskite solar cells and frontiers in the era of PSC. Firstly, the review begins by discussing the evolution of different generations of solar cells and the development of PSC. Next, the working mechanism of the PSC system, distinctive optical properties, and use of interfacial engineering are discussed.

Secondly, some effective approaches to enhance the output of PSCs are summarised. Appropriate ETL and HTL can spontaneously reduce interface electrical and optical losses. Carbon, $NiO_x$, Au, and Cu can be the best options to promote the commercial application of PSC. Moreover, structure modulation of PSC to modify Eg and interface morphology may be employed to enhance the PCE.

The use of the effective solvent additive in preparing the PSC absorber required great attention to improve the morphology and crystallinity of the perovskite films. Moreover, it helps to present low trap states and defects in order to allow for a long carrier lifetime by tuning the optical and electrical properties. The type of electrode used can also affect the PSC's stability, which can be resolved by using alternate electrodes, such as transparent electrodes.

Thirdly, besides all the development in PSCs, most studies concluded that the stability of devices is still a matter of concern. The main factors affecting the stability of PSCs involve UV light, moisture, temperature, humidity, water, and oxygen. It is essential to prevent perovskite films from the degradation mechanism to protect the cell's lifetime

and make it favourable for commercialisation. The results of various researchers on I-V hysteresis in perovskite material were revised. Hysteresis plays a crucial role in the accurate determination of PCE of PSC. Various theories present different causes, e.g., ferroelectric, ion migration, charge trapping, and the capacitive effect on which hysteresis is dependent. However, many fundamental questions about the mechanism of hysteresis and its causes need to be answered.

Moreover, the exciting idea of using perovskite solar cells with concentrated light is discussed. Concentrated Per/Si tandem solar cells as a single junction or multijunction compared to other rival PV technologies (thin-film, organic, dye-sensitised) could be the best alternative solar absorber. However, a key factor in limiting the PCE of concentrated perovskite solar cells could be possible due to the deterioration of FF at a high light intensity. However, the reduction of series resistance in the charge extraction layer can cope with this issue. Finally, some new strategies need to be developed in order for it to operate at low temperatures under a high level of irradiance, as well as new optical designs to capture the maximum irradiance for the viability of PSC systems. Nonetheless, there is still enough room for more development in order to garner an understanding of fabricating low-cost perovskite solar cells.

Finally, the cost analysis of previous systems concludes that PSC systems have more market shares than competitors with high costs. Therefore, new research is needed in order to fabricate high-quality, more stable, and high-output PSCs to compete with other PV technologies. At the same time, some boards are not explained in depth and need to be further explored. The continuous research and advancement in the perovskite field with material development will help boost the technology and present better elucidation in future.

**Funding:** This work was partly funded by the Engineering and Physical Science Research Council through the End User Energy Demand technology project (EP/S030786/1).

**Conflicts of Interest:** The authors declare no conflict of interest.

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
