# Peer review of "Stability and Performance Enhancement of Perovskite Solar Cells: A Review"

_energies, doi:10.3390/en16104031_

Round 1

Reviewer 1 Report

Comments to the Author

In this work, the authors reviewed on “Stability and performance enhancement of Perovskite solar cells using optical concentration: A Review”. This work is well presented through figures and is beneficial to new readers. This work is well suited to this journal after minor revisions.

1. Please check the updated efficiency and add the correct value.

2. All grammatical errors and typos should be removed to make clear and meaningful sentences.

3. Authors can add other additives-based reports too.

4. Current trend is fabricating tandem perovskite solar cells by thermal evaporation technique for the large-area based devices. So, it’s better to add thermal evaporation-based perovskite solar cells reports.

5. Authors should carefully check all equations. 

Author Response

  1. Please check the updated efficiency and add the correct value.

Answer: The updated efficiency of the PSC devices has been added throughout the manuscript in all the sections.

  1. All grammatical errors and typos should be removed to make clear and meaningful sentences.

      Answer: Sorry for any inaccuracy; we have eliminated the issue in the revised manuscript.

  1. Authors can add other additives-based reports too.

      Answer: We have added other additive-based reports on page 23-24.

  1. Current trend is fabricating tandem perovskite solar cells by thermal evaporation technique for the large-area based devices. So, it’s better to add thermal evaporation-based perovskite solar cells reports.

      Answer: Thermal evaporation technique has been added to the manuscript (page 23-24)

  1. Authors should carefully check all equations. 

      Answer: All the equations have been revised and corrected.

Author Response

This paper has reviewed the photovoltaic performance of perovskite solar cells (PSC). However, the title In the whole manuscript I did not find any aspect of the optical concentration, which to me is concentration of the solar energy making it more than one sun. In section 3. Pathways perovskite solar cells, refers to the use of multijunction structures for enhancing the power conversion efficiency (PCE), which is not optical concentration. Therefore, the title should be changed to reflect the actual contents of the manuscript solar cells: A review The stability aspect is not fully covered, I believe, because nowhere is says what is the current status of stability achieved in perovskite solar cells (PSCs). I have added some more in my comments below. Therefore, the review in its present form provides no new information to the readers and not acceptable. In addition, authors should also address the following issues before this paper can be processed any further.

Answer: title of the manuscript has been changed to "Stability and performance enhancement of Perovskite Solar Cells: A Review". Likewise in Section 3 title is changed to. Some new trends and current status and the stability of the perovskite has been added in the manuscript.

  1. Authors need to add more references in :

Answer: All the references have been added.

  1. Line 142, In 2006, Japanese researchers.,… [11]. However, the ref . 11 in the list is 1994 paper. Please cheque and correct it.

       Answer: The correct reference has been added.

  1. Lines 175- does not make any sense, please rewrite

Answer: The sentence has been rewritten.

The decomposition of MAPbI3 at a lower temperature does not allow be used commercially [1].

  1. Line 186, “protons” should be changed to “photons”.

Answer: The modification has been made.

  1. Line 215-216, “Though F- exhibit….. other halogens”. This sentence is not complete. Please rewrite.

Answer: the sentence has been rewritten.

Though F- exhibit excellent robust bonding with H atom of A site and B site compared to other halogens.

  1. Figure captions of Fig. 5, needs to be expanded to explain the contents of Fig. 5. It is not clear where to start in this figure.

Answer:

  1. Lines 256 – “The enhanced … scope”. Rewrite the sentence. It does not make sense.

Answer: The sentence has been rewritten.

The enhanced PCE of this perovskite material compared to MA+ engaged to the higher absorption rate due to better film quality, fewer pin-holes, higher crystal quality, larger grain sizes, smaller roughness within the grains.

  1. Line 299 , “ the device… turned as well”. Please rewrite the sentence, it doesn't make any sense.

Answer: The sentence has been rewritten.

Generally, Sn2+ metal-based perovskite devices presented lower bandgap as compared to Pb2+-based devices influencing the stability of the device and reduced PCE.

  1. Section 2.4 Stability studies of PSC. Please improve this section by clearly adding what is the current status on the stability of PSCs, 2 days, 2 years or 5 hours?

         Answer:    Latest literature has been added in the stability section with time period.

  1. but they are expected-  that all panels must be stable for 25 years.Please rewrite this.

Answer: Modification has been made to the statement.

      The predicted steady state performance and stability of standard PV modules is 20 to 25 years. However, depending on various characteristics and properties of the com-ponents of PV devices such stability has not demonstrated in the last few years.

  1. There are many more such errors which need to be carefully checked and

Answer: The manuscript has been revised. Carefully to correct all the typos and grammatical mistakes.

  1. Finally, the English language used requires serious

Answer: The Manuscript has been revised to improve the English language.